# JOINT TRAINING DOES NOT TRANSFER INFORMATION BETWEEN EEG AND IMAGE CLASSIFIERS

## ABSTRACT

Caution is necessary with machine-learning methods, and especially computer-vision methods, to support brain processing claims from neuroimaging data. Recent papers propose (i) a joint-training process that does not use class information and (ii) a bidirectional transfer of (a) image information to an EEG classifier and (b) brain-activity information to an image classifier, such that the joint embedding includes the shared image and brain-activity information. These claims cannot be maintained: the training process is initialized with class information, and joint training with EEG degrades rather than improves the performance of the image encoder. Moreover, theoretical solutions exist that entail no transfer beyond class information in the joint embedding space.

## 1 INTRODUCTION

Recent papers (Palazzo et al., 2018; 2020a;b; 2021) propose a method for jointly training two neural networks, one to map images to encodings and the other to map EEG data from subjects viewing those images to encodings, so that the encodings for the images are similar to those from the EEG data. Specifically, they claim that

  I) *class labels are not used anywhere in the equation. This makes sure that the resulting embedding does not just associate class discriminative vectors to EEG and images, but tries to extract more comprehensive patterns that explain the relations between the two data modalities.* (Palazzo et al., 2021, § 3 ¶ 5) and

  II) *our multimodal approach learns a joint brain-visual embedding and finds similarities between brain representations and visual features* (Palazzo et al., 2021, § 1 ¶ 2).

They use the resulting encodings for EEG classification, image classification, saliency detection, and producing activation maps that decode brain representations. Central are the additional implicit claims

 III) that the joint embedding space encodes both information about the images and subjects' brain activity from viewing the images and

 IV) that the joint-training process bidirectionally transfers information from the image encoders to the EEG encoders, and information from the EEG encoders to the image encoders.

Other work (Li et al., 2021; Ahmed et al., 2021; 2022; Bharadwaj et al., 2023) questioned these claims due to confounds in the data (Spampinato et al., 2017) used by Palazzo et al. (2018; 2020a;b; 2021), which was collected with a block design with all stimuli of a given class presented to subjects in close proximity and the training and test sets containing samples from the same block. Since EEG data contains temporal drift, this drift is classified rather than stimulus-related brain activity. Li et al. (2021), Ahmed et al. (2021; 2022), and Bharadwaj et al. (2023) note that it is critical to remove the confound by breaking the correlation between temporal drift and stimulus class by randomizing stimulus presentation order. Li et al. (2021), Ahmed et al. (2021; 2022), and Bharadwaj et al. (2023) demonstrated that numerous classifiers, including the one used in Palazzo et al. (2018; 2020a;b; 2021), lose high classification accuracy seen on the data from Spampinato et al. (2017) used in Palazzo et al. (2018; 2020a;b; 2021), dropping to chance with nonconfounded data properly collected from randomized trials.

Here, we focus on a separate issue. We demonstrate that their method itself does not exhibit claims I–IV, whether applied to confounded or nonconfounded data. We contribute the novel observations

that independently refute the claims in Palazzo et al. (2018; 2020a;b; 2021) beyond Li et al. (2021), Ahmed et al. (2021; 2022), and Bharadwaj et al. (2023):

A) Prior to joint training, the pretrained image encoders contain close to perfect class information, and likely very little information other than class information. This calls claim I above into question.

B) The models used as EEG and image encoders, and the loss function used to jointly train those encoders, have the representational capacity to memorize one-hot EEG and image encodings that minimize the loss function on the training set. These models can be found with separate training. Thus it is unlikely that joint training transfers information from the image encoders to the EEG encoders or vice versa, or includes anything other than class information. This calls claims II–IV above into question.

We demonstrate that their method doesn't support their claims, even when applied to data that remedies the confound. Since their claims are contingent upon production of joint embeddings that include both image and brain-activity information, this calls all claims in Palazzo et al. (2018; 2020a;b; 2021) into question.

Beyond this, we further demonstrate

C) that joint training appears to (1) use the class information included in the pretrained image encoders to train the EEG encoder as a classifier, but (2) otherwise degrades performance of the image encoder as a classifier and

D) that while the EEG encoders so trained can perform above chance on confounded data, they critically perform at chance on nonconfounded data.

Thus their joint-training method is ineffective.

Here, we limit our analyses and discussion to claims A–D, in so far as they refute claims I–IV. This does not hinge on the confound reported in Li et al. (2021), Ahmed et al. (2021; 2022), and Bharadwaj et al. (2023); in fact, we perform the same analyses both on confounded and nonconfounded data, obtaining the same results. Our logical arguments all are based on the information content in the encodings at various stages of various kinds of training, where information content is measured by principal component analysis (PCA). It is expressly not based on absolute classification accuracy, *i.e.*, how well the classifiers perform. We are concerned with relative classification accuracy of various reconstructions of the encodings with limited sets of components derived through PCA as a way of assessing information content of the encodings.

Palazzo et al. (2018; 2020a;b; 2021) use the EEG and image encodings produced by jointly trained EEG and image encoders to produce saliency and activation maps. The validity of these maps hinges on the claim that these encodings contain both visual and brain-activity information. We demonstrate this to be false, calling the validity of these maps into question. Li et al. (2021, § 5.4) already questioned this validity both on methodological grounds and as a result of the confound. Our results here offer additional independent reason to question the validity of these maps.

We make no claim that it is difficult or impossible to perform the central task attempted by Palazzo et al. (2018; 2020a;b; 2021), namely to perform joint training of EEG and image encoders to transfer image information to the EEG encoder and brain-activity information to the image encoder and yield a joint embedding that contains both image and brain-activity information, without providing class information. It might be possible to perform this task with a different method, or even with the same method but with different data. Our sole claim is that their experiments and analyses do not demonstrate that they have performed this central task as stated by claims I–IV made in Palazzo et al. (2018; 2020a;b; 2021).

## 2 SIGNIFICANCE

Several independent lines of research have refuted a large body of flawed work (Spampinato et al., 2017; Kavasidis et al., 2017; Palazzo et al., 2017; 2018; 2020a;b; 2021; 2024) along completely different axes. Li et al. (2021) demonstrated that the dataset used (Spampinato et al., 2017), and the methods used to collect that dataset, suffer from a temporal confound, correlating stimulus class with experiment timing. Accuracy drops to chance when the confound is removed. Ahmed et al. (2021) demonstrated that this holds even with a much larger dataset. Ahmed et al. (2022) demonstrated that

this holds for the additional classifiers used in Palazzo et al. (2018; 2020a;b; 2021). Bharadwaj et al. (2023) demonstrated that this holds even when using supertrials.

Here we progress beyond prior demonstrations that the particular dataset (Spampinato et al., 2017) is confounded. We offer the novel claim that the joint training regimen is also flawed. This is significant because the same joint training regimen continues to be used (Bai et al., 2023; Zeng et al., 2023b; Ahmadieh et al., 2023; Lan et al., 2023; Du et al., 2023; Liu et al., 2023; Song et al., 2023; Ye et al., 2024). None of the work that jointly trains EEG and image classifiers even attempts to assess whether joint training is effective in information transfer. In most cases, we can't perform the analysis because the published work does not contain sufficient information to do so, including but not limited to stimulus timing and presentation order. Without this, one cannot even be sure that the data is free from confounds.

This is further significant for the following reasons:

- Nearly one hundred papers (An & Cho, 2016; Spampinato et al., 2016; Ben Said et al., 2017; Bozal Chaves, 2017; Kavasidis et al., 2017; Palazzo et al., 2017; Parekh et al., 2017; Spampinato et al., 2017; Zhang et al., 2017; Du et al., 2018; Fares et al., 2018; Kumar et al., 2018; Palazzo et al., 2018; Piplani et al., 2018; Tirupattur et al., 2018; Wang et al., 2018; Zhang & Liu, 2018; Zhang et al., 2018; Zhong et al., 2018; Du et al., 2019; Hwang et al., 2019; Jiang et al., 2019; Jiao et al., 2019; Long et al., 2019; Mukherjee et al., 2019; Uys, 2019; Wang et al., 2019; Cudlenco et al., 2020; Fares et al., 2020; Li et al., 2020; Palazzo et al., 2020a;b; Wang et al., 2020; Zheng et al., 2020a;b; Palazzo et al., 2021; Zheng & Chen, 2021; Ma et al., 2021; Mo et al., 2021; Jiang et al., 2021; Lee et al., 2021; Cavazza et al., 2022; Khaleghi et al., 2022; Lee et al., 2022; Mishra et al., 2022; Mishra, 2022; Scharnagl & Groth, 2022; Shimizu & Srinivasan, 2022; Ahmadieh et al., 2023; Bai et al., 2023; Du et al., 2023; Duan et al., 2023; Hasan & A, 2023; Imani et al., 2023; Lan et al., 2023; Lee et al., 2023; Liu et al., 2023; Singh et al., 2023; Song et al., 2023; Wahengbam et al., 2023; Zeng et al., 2023b;a; Fan et al., 2024; Ferrante et al., 2024a;b; Gou et al., 2024; Lei et al., 2024; Liu et al., 2024a;b; Luvsansambuu et al., 2024; Mishra et al., 2024; Mwata-Velu et al., 2024; Ngo et al., 2024; Palazzo et al., 2024; Pan et al., 2024; Qian et al., 2024; Singh et al., 2024; Tang et al., 2024; de la Torre-Ortiz et al., 2024; Yang & Liu, 2024; Ye et al., 2024; Zheng et al., 2024b;a; Zhu et al., 2024; Deng et al., 2025; Fares, 2025; Fu et al., 2025; Lopez et al., 2025; Mehmood et al., 2025; Singh et al., 2025; Xiang et al., 2025) draw flawed conclusions based on the confounded dataset from Spampinato et al. (2017) and datasets suffering from the same confound.
- A number of new datasets have been collected with this same confounded protocol (Gou et al., 2024; Pan et al., 2024; Zhu et al., 2024; Qian et al., 2024; Uys, 2019; Shimizu & Srinivasan, 2022; Liu et al., 2024b; Wang et al., 2019; 2020; Ma et al., 2021; Cudlenco et al., 2020; Zheng et al., 2024b; Cavazza et al., 2022; Luvsansambuu et al., 2024; Liu et al., 2023; Bai et al., 2023; Parekh et al., 2017).
- A number of these have been publicly released and are used by others. For example, Singh et al. (2023), Singh et al. (2024), and Lopez et al. (2025) use the dataset reported in Kumar et al. (2018) and Duan et al. (2023), Singh et al. (2024), and Lopez et al. (2025) use the dataset reported in Ma et al. (2021).
- This is further egregious because Palazzo et al. (2020b; 2024) continue to claim that their dataset (Spampinato et al., 2017), and their results that were obtained with that dataset (Spampinato et al., 2017; Kavasidis et al., 2017; Palazzo et al., 2017; 2018; 2020a;b; 2021; 2024), are valid, despite the refutations in Li et al. (2021), Ahmed et al. (2021; 2022), and Bharadwaj et al. (2023), in part, because of claims I–IV in Palazzo et al. (2018; 2020a;b; 2021).
- This has been used to justify continued publication of a large and growing body of flawed work based on confounded datasets (Cavazza et al., 2022; Khaleghi et al., 2022; Lee et al., 2022; Mishra et al., 2022; Mishra, 2022; Scharnagl & Groth, 2022; Shimizu & Srinivasan, 2022; Ahmadieh et al., 2023; Bai et al., 2023; Du et al., 2023; Duan et al., 2023; Hasan & A, 2023; Imani et al., 2023; Lan et al., 2023; Lee et al., 2023; Liu et al., 2023; Singh et al., 2023; Song et al., 2023; Wahengbam et al., 2023; Zeng et al., 2023b;a; Fan et al., 2024; Ferrante et al., 2024a;b; Gou et al., 2024; Lei et al., 2024; Liu et al., 2024a;b; Luvsansambuu et al., 2024; Mishra et al., 2024; Mwata-Velu et al., 2024; Ngo et al., 2024; Palazzo et al., 2024; Pan et al., 2024; Qian et al., 2024; Singh et al., 2024; Tang et al., 2024; de la

Torre-Ortiz et al., 2024; Yang & Liu, 2024; Ye et al., 2024; Zheng et al., 2024b;a; Zhu et al., 2024; Deng et al., 2025; Fares, 2025; Fu et al., 2025; Lopez et al., 2025; Mehmood et al., 2025; Singh et al., 2025; Xiang et al., 2025) even after the confound became known through the work of Li et al. (2021), Ahmed et al. (2021; 2022), and Bharadwaj et al. (2023).

Current machine-learning conferences, and more generally, computer-science conferences and journals, are loathe to publish refutations. Observing this, Schaeffer et al. (2025) proposed that the field of machine-learning establish a "refutations and critiques" track in prominent conferences. While we applaud and support this proposal, the current lack of such a track should not be an impediment to publishing refutations. Scientific journals in other fields have long done so, often resulting in retraction of flawed work. Schaeffer et al. (2025) offer five example pieces of claimed flawed work in machine learning. Each is an individual paper. These pale in comparison to the flaws we uncover here: a systemic flaw of the entire peer review process across an entire field of inquiry, namely classification of stimulus image class from EEG recordings, that affects seventeen datasets and ninety one papers. Moreover, none of the five examples in Schaeffer et al. (2025) are egregious; here the authors of the flawed work continue to argue for its validity despite four refereed refutations and fifty new flawed papers have been published subsequent to these four refereed refutations. This argues for the need to make the community aware of the severity of the issue.

## 3 METHOD

We jointly train the EEG encoder (EEGChannelNet, Palazzo et al., 2018; 2020a;b; 2021) with each of the same image encoders employed by Palazzo et al. (2018; 2020a;b; 2021): Inception v3 (Szegedy et al., 2016), ResNet-101 (He et al., 2016), DenseNet-161 (Huang et al., 2017), and AlexNet (Krizhevsky et al., 2012). Our analyses report results for all four image encoders, each analyzed two ways: without pretraining ('no pretraining') and pretrained on the ILSVRC 2012 training set ('pretraining'). As we evaluate just the encodings, we do not postpend classifiers to these encoders. Thus these encoders produce 1000-element output vectors, not 40-element ones.

We train the encoders in two ways: (1) joint training as described in Palazzo et al. (2018; 2020a;b; 2021), with triplet loss, and (2) separate training individually with MSE loss against one-hot class encodings. After separate training, we run a single epoch of joint training, without updating the weights, just to compute the triplet loss. We report three kinds of results: 'before' is before any joint or separate training, 'after joint' is after joint training, and 'after separate' is after separate training.

We repeat this analysis for three different EEG datasets: 'Spampinato et al. (2017)' refers to the data from Spampinato et al. (2017) used by Palazzo et al. (2018; 2020a;b; 2021), 'Li et al. (2021) block' refers to subject 6 first image block run from Li et al. (2021), and 'Li et al. (2021) randomized' refers to subject 6 image rapid-event run from Li et al. (2021). 'Spampinato et al. (2017)' and 'Li et al. (2021) block' are confounded; 'Li et al. (2021) randomized' is not.

We use the encoders, trained just on the training sets, to produce encodings on the validation and test sets for all splits and pool the results. For the 'Li et al. (2021) block' and 'Li et al. (2021) randomized' data, the validation and test sets are a disjoint cover of the dataset so each sample has exactly one encoding. Critically, the data from Spampinato et al. (2017) does not have this property. We perform two analyses on these encodings. First, we perform principal-component analysis (PCA) and reconstruct the encodings two ways: one with just the 'top 40' components and one with just the 'bottom 960' components and report the fraction of variance in the encodings explained by the top 40 components. Second, for both the raw encodings ('original') and the reconstructed encodings ('top 40' and 'bottom 960'), we compute the accuracy when using the encodings for classification, without postpending any classifier (a 1000→40 FC layer followed by a softmax as done by Palazzo et al. 2018; 2020a;b; 2021). Since the 40 classes employed by Palazzo et al. (2018; 2020a;b; 2021) are a subset of the 1000 classes in ILSVRC 2012, and the image stimuli used by Spampinato et al. (2017) to elicit EEG response are a subset of the training images in ILSVRC 2012, one can read off the class label directly from the 1000-element encoding produced by either the EEG or image encoders by choosing the index of the maximal element. We do this two ways. The first ('1000 classes') considers maximal elements outside of the 40 classes to be misclassification. The second ('40 classes') ignores elements outside of the 40 classes and only computes the maximal element among the 40 classes.

We also analyze the value of the loss function: for the triplet loss after both 'joint' and 'separate' training, on all four image encoders, both with and without 'pretraining,' and all three datasets, and also the MSE loss when separately training the 'EEG' and 'image' encoders. These losses are computed per sample and per element of the 1000-element encoding, averaged over samples and splits.

The appendix contains more details of our method.

## 4 RESULTS

Table 1 shows the fraction of variance explained by the top 40 principal components of the encodings produced for both modalities ('EEG' and 'image') for all image encoders and datasets, 'before' joint or separate training, 'after joint' training, both with and without 'pretraining,' and 'after separate' training.[1]

- The top 40 principal components explain a large portion of the image-encoding variance ($\geq$63.9%) before training (column v; Table 5[2]). This together with Table 2 implies that the pretrained image encoders produce encodings that contain (primarily) class information on all data. This supports claim A and calls claim I into question.
- After joint training with pretraining on the 'Spampinato et al. (2017)' data, the top 40 principal components explain almost all ($\geq$94.9%) of the variance in the image encodings (column vi, for rows i–iv; Table 6). This implies that the image encoders jointly trained on the 'Spampinato et al. (2017)' data produce encodings with little more than class information.
- After separate training, the top 40 principal components explain almost all of the variance ($\geq$96.8%) in the image encodings (column viii; Table 7). This implies that the separately trained image encoders produce encodings with little more than class information on all data.
- After separate training, the top 40 principal components explain almost all of the variance ($\geq$96.8%) in both the EEG and image encodings on the 'Spampinato et al. (2017)' data (columns iv and viii, for rows i–iv; Table 8). This implies that the separately trained EEG and image encoders produce encodings with little more than class information on the 'Spampinato et al. (2017)' data.

Collectively this suggests that after either joint or separate training the image encodings contain class information and mostly class information. Likewise, after separate training on the 'Spampinato et al. (2017)' data, the EEG encodings contain primarily class information. This is not surprising, as has been noted, the 'Spampinato et al. (2017)' data is confounded. This supports claim B and calls claims II–IV into question.

Table 2 shows the accuracy of classifying the encodings of both modalities ('EEG' and 'image') for all image encoders and datasets, 'before' joint or separate training, 'after joint' training, both with and without 'pretraining,' and 'after separate' training, using the raw encodings ('original') or the encodings reconstructed from the 'top 40' or 'bottom 960' principal components, both when considering all ILSVRC 2012 classes ('1000 classes') and when only considering the '40 classes' used by Palazzo et al. (2018; 2020a;b; 2021).

- Image classification accuracy is near perfect with all components with 40 classes ($\geq$95.5%; column i, for rows v–viii, xiii–xvi, and xxi–xxiv; Table 9) and very high with 1000 classes ($\geq$77.9%; column xiii, for rows v–viii, xiii–xvi, and xxi–xxiv; Table 10) before training. Likewise with just the top 40 principal components ($\geq$ 85.7% and $\geq$72.2%; columns v and xvii, for rows v–viii, xiii–xvi, and xxi–xxiv; Table 11). This holds both for the 'Spampinato et al. (2017)' data and the 'Li et al. (2021) block' and 'Li et al. (2021) randomized' data. This demonstrates that they are giving class information implicitly to their training

---

[1]Note that while the entries in Table 1 are percentages, they are not classification accuracies. Explained variance measures how close a reconstruction of the encodings using just the top 40 components is to the original encodings. High values indicate that the reconstruction is close to the original encodings because the bulk of the information in the encodings resides in the top 40 components.

[2]Here and throughout, tables numbered five and higher refer to those in the appendix. These contain variants of the five main tables with the indicated columns and rows highlighted in color.

Table 1: Explained variance (%) in the top 40 principal components of the encodings, 'before' joint or separate training, 'after joint' training, both with and without 'pretraining,' and 'after separate' training.

| | | | EEG before | EEG after joint pretraining | EEG after joint no pretraining | EEG after separate | image before | image after joint pretraining | image after joint no pretraining | image after separate |
|---|---|---|---|---|---|---|---|---|---|---|
| | | | i | ii | iii | iv | v | vi | vii | viii |
| Spampinato et al. (2017) | Inception v3 | i | 44.7 | 37.2 | 15.3 | 99.5 | 64.1 | 94.9 | 98.9 | 100.0 |
| | ResNet-101 | ii | 46.5 | 36.2 | 17.7 | 99.5 | 82.0 | 98.9 | 98.2 | 96.8 |
| | DenseNet-161 | iii | 48.2 | 25.6 | 22.7 | 100.0 | 75.4 | 97.4 | 90.2 | 100.0 |
| | AlexNet | iv | 46.3 | 21.7 | 18.5 | 99.5 | 82.3 | nan | 100.0 | 99.7 |
| Li et al. (2021) block | Inception v3 | v | 83.3 | 75.9 | 69.4 | 42.3 | 63.9 | 57.4 | 98.3 | 99.9 |
| | ResNet-101 | vi | 85.4 | 79.8 | 72.8 | 41.9 | 81.9 | 99.5 | 99.0 | 99.7 |
| | DenseNet-161 | vii | 84.0 | 79.3 | 76.3 | 42.9 | 75.3 | 69.4 | 94.3 | 99.5 |
| | AlexNet | viii | 84.4 | 78.9 | 73.2 | 40.0 | 82.2 | 78.9 | 99.9 | 98.9 |
| Li et al. (2021) randomized | Inception v3 | ix | 62.1 | 40.5 | 38.3 | 19.3 | 63.9 | 60.2 | 98.1 | 99.9 |
| | ResNet-101 | x | 61.9 | 52.2 | 43.6 | 19.7 | 81.9 | 99.1 | 98.8 | 99.8 |
| | DenseNet-161 | xi | 63.5 | 48.5 | 46.5 | 19.6 | 75.3 | 70.8 | 93.9 | 99.7 |
| | AlexNet | xii | 63.6 | 49.5 | 41.3 | 19.2 | 82.2 | 86.9 | 99.9 | 98.8 |

by use of image encoders pretrained on ImageNet, supporting claim A, and calling claim I into question.

- Image classification accuracy is much lower ($\leq 46.0\%$) before training when discarding the top 40 principal components (columns ix and xxi, for rows v–viii, xiii–xvi, and xxi–xxiv; Table 12). This holds both for 40 classes and 1000 classes and both for the 'Spampinato et al. (2017)' data and the 'Li et al. (2021) block' and 'Li et al. (2021) randomized' data. Thus there is much less class information outside the top 40 principal components. This demonstrates that they are giving little more than class information implicitly to their training by use of image encoders pretrained on ImageNet, further supporting claim A and calling claim I into question.

- With only five exceptions, all in the bottom 960 principal components, image classification accuracy decreases after joint training with pretraining (compare columns ii, vi, x, xiv, xviii, and xxii to i, v, ix, xiii, xvii, and xxi, respectively, for rows v–viii, xiii–xvi, and xxi–xxiv; Table 13). This holds both for 40 classes and 1000 classes, both for the 'Spampinato et al. (2017)' data and the 'Li et al. (2021) block' and 'Li et al. (2021) randomized' data, and whether using all components or just the 40 principal components. This suggests that joint training is hurting, not helping, supporting claim C(2).

- With the exception of a small number of cases that are marginally above chance, EEG classification accuracy is at chance except after separate training on confounded data either on all components or the top 40 components (columns iv, viii, xvi, and xx, for rows i–iv and ix–xii; Table 14). It is also above chance after separate training on the 'Li et al. (2021) block' data in the bottom 960 components (columns xii and xxiv, for rows ix—xii; Table 15). This suggests that separate training is able to extract class from confounded data but not nonconfounded data and that joint training is not able to extract class from any data, supporting claims C(1) and D.

- While the image encoder can sometimes generalize (some of columns ii, iv, vi, viii, x, xii, xiv, xvi, xviii, xx, xxii, and xxiv, for rows v–viii, xiii–xvi, and xxi–xxiv, are above chance; Table 16) and the EEG encoder can sometimes generalize on confounded data with separate training (some of columns iv, viii, xii, xvi, xx, and xxiv, for rows i–iv and ix–xii, are above chance; Table 17), the EEG encoder cannot generalize on nonconfounded data with joint training with pretraining (all of columns ii, vi, x, xiv, xviii, and xxi, for rows xvii-xx, are at chance; Table 18). This supports claim D.

- Classification accuracy is at chance after joint training without pretraining (columns iii vii, xi, xv, xix, and xxiii; Table 19). This holds both for EEG classification and image classification, both for 40 classes and 1000 classes, both for the 'Spampinato et al. (2017)' data and the 'Li et al. (2021) block' and 'Li et al. (2021) randomized' data, and whether using all components, just the 40 principal components, or the bottom 960 components. This suggests that their method completely breaks down when not provided with class information through pretraining, and further supports claim A and calls claim I into question.

The results in Table 2 exhibit some differences between different image encoders for each analysis ('EEG' *vs.* 'image,' '40 classes' *vs.* '1000 classes,' 'original' *vs.* 'top 40' *vs.* 'bottom 960,' and 'before' *vs.* 'after joint' *vs.* 'after separate'). This is not surprising. Different image classifiers exhibit

different classification accuracy, especially when pretrained with different training regimens. It is further not surprising that this difference manifests with different PCA reconstructions with different components and also manifests when subjected to joint fine tuning with noisy EEG data or separate fine tuning on the tiny subset of ImageNet used here, because fine tuning can break different models in different ways. None of these differences impact our claims A–D or the refutation of claims I–IV from Palazzo et al. (2018; 2020a;b; 2021).

Table 3 reports the per-sample triplet loss on the training set, after 'joint' training, both with and without 'pretraining,' and 'separate' training, for all image encoders and datasets.

- With only two exceptions, separate training gets to a lower loss than joint training with pretraining (compare columns vii, viii, and ix to i, ii, and iii, respectively; Table 20). This critically suggests that there is a point within the representational-capacity space of their model and loss function that achieves lower loss than achieved by their joint-training procedure. The joint-training procedure could have achieved it, it just didn't. The fact that point was achieved with separate training, indicates that the resulting EEG encodings do not have any image information and the resulting image encodings do not have any brain-activity information. This supports claim B and calls claims II–IV into question.
- With only three exceptions, joint training without pretraining gets to a lower loss than with pretraining (compare columns iv, v, and vi to i, ii, and iii, respectively; Table 21). Yet classification accuracy is at chance after joint training without pretraining (columns iii vii, xi, xv, xix, and xxiii of Table 2; Table 19). This suggests that joint training without pretraining can memorize the training set yet fail to generalize at all, and further supports claim A and calls claim I into question.

Table 4 reports the per-sample MSE loss of the individual encoders of both modalities ('EEG' and 'image') on the training set, after separate training, for all image encoders and datasets.

- Separate training against one-hot labels can get to very low MSE (all columns and rows), indicating that the model has the representational capacity to memorize both the EEG and image training data. Achieving this point in the representational-capacity space with separate training and the encodings being one-hot implies that at this point, the EEG and image encodings have nothing but class information. Since this point could have been achieved with joint training, joint training could have produced a set of encodings that have nothing but class information. This supports claim B and calls claims II–IV into question.

Collectively our results suggest that:

- Their statement (claim I) about not incorporating any class information in joint training is false (our claim A).
- Their encoder models combined with their triplet loss function allow a point in the space where the encodings on the training set are one-hot and thus cannot possibly contain anything other than class information. This point has zero triplet loss and thus is the minimum. It is achievable. Thus their framework can produce encodings with nothing but class information.
- This point can be reached by an alternate separate-training regimen that clearly does not transfer any information between the EEG and image encoders.

It is highly unlikely that the suboptimal EEG model that they produce by joint training has any image information, the suboptimal image model that they produce by joint training has any brain-activity information, and neither suboptimal model produced by joint training has anything other than class information, even when run on nonconfounded data. This supports claim B and calls claims II–IV into question.

## 5 CONCLUSION

We implore all future EEG image classification effort to release **raw data** that includes stimulus timing and presentation order, not preprocessed data. We further implore all future effort to jointly train EEG and image classifiers to employ the methods presented here to assess the effectiveness of purported information transfer attributed to joint training, and not to assume such effectiveness due to classification accuracy and image reconstruction.

Table 2: Original classification accuracy and classification accuracy after reconstruction (%), using just the encodings, 'before' joint or separate training, 'after' joint training, both with and without 'pretraining,' and 'after separate' training.

Column groups (each group: *before* | *after joint* — pretraining, no pretraining | *after separate*):

- **40 classes** — original (i–iv), top 40 (v–viii), bottom 960 (ix–xii)
- **1000 classes** — original (xiii–xvi), top 40 (xvii–xx), bottom 960 (xxi–xxiv)

| Dataset | Mod. | Model | i | ii | iii | iv | v | vi | vii | viii | ix | x | xi | xii | xiii | xiv | xv | xvi | xvii | xviii | xix | xx | xxi | xxii | xxiii | xxiv |
|---|---|---|---|---|---|---|---|---|---|---|---|---|---|---|---|---|---|---|---|---|---|---|---|---|---|---|
| Spampinato et al. (2017) | EEG | Inception-v3 | 2.6 | 3.0 | 2.5 | 55.5 | 2.2 | 2.6 | 2.5 | 55.5 | 2.5 | 3.1 | 2.5 | 2.3 | 0.1 | 0.1 | 0.1 | 55.5 | 0.1 | 0.1 | 0.1 | 55.5 | 0.1 | 0.1 | 0.1 | 2.3 |
| | | ResNet-101 | 2.3 | 3.0 | 2.6 | 54.7 | 2.8 | 2.4 | 2.4 | 54.7 | 2.4 | 3.1 | 2.4 | 2.3 | 0.0 | 0.1 | 0.1 | 54.7 | 0.0 | 0.0 | 0.0 | 54.7 | 0.1 | 0.1 | 0.0 | 2.3 |
| | | DenseNet-161 | 2.5 | 3.1 | 2.4 | 57.4 | 2.4 | 2.6 | 2.3 | 57.4 | 2.5 | 3.2 | 2.4 | 2.5 | 0.1 | 0.1 | 0.1 | 57.4 | 0.0 | 0.1 | 0.0 | 57.4 | 0.1 | 0.1 | 0.1 | 2.5 |
| | | AlexNet | 2.3 | 2.5 | 2.4 | 57.3 | 2.4 | 2.6 | 2.6 | 57.3 | 2.4 | 2.5 | 2.6 | 2.5 | 0.1 | 0.1 | 0.1 | 57.3 | 0.0 | 0.1 | 0.0 | 57.3 | 0.1 | 0.1 | 0.1 | 2.5 |
| | image | Inception-v3 | 99.7 | 43.2 | 2.7 | 3.4 | 99.4 | 13.5 | 2.9 | 2.2 | 45.5 | 29.4 | 2.7 | 2.5 | 92.8 | 16.5 | 0.0 | 0.0 | 98.9 | 4.0 | 0.0 | 0.0 | 12.8 | 9.3 | 0.0 | 0.0 |
| | | ResNet-101 | 99.4 | 24.2 | 2.4 | 82.5 | 98.3 | 7.3 | 2.9 | 70.5 | 33.0 | 9.5 | 2.1 | 39.4 | 90.5 | 2.6 | 0.0 | 82.5 | 95.6 | 0.3 | 0.0 | 70.5 | 9.5 | 0.7 | 0.0 | 39.2 |
| | | DenseNet-161 | 99.2 | 7.2 | 2.4 | 23.3 | 97.9 | 4.4 | 2.3 | 3.5 | 42.7 | 3.8 | 2.6 | 2.6 | 89.4 | 0.1 | 0.0 | 12.1 | 95.8 | 0.0 | 0.0 | 0.1 | 14.2 | 0.0 | 0.1 | 0.0 |
| | | AlexNet | 95.8 | 2.6 | 2.7 | 18.7 | 85.8 | 2.6 | 2.7 | 19.1 | 34.1 | 2.6 | 2.4 | 2.4 | 78.4 | 0.0 | 0.3 | 18.7 | 72.5 | 0.0 | 0.0 | 19.1 | 12.5 | 0.0 | 0.0 | 2.4 |
| Li et al. (2021) Block | EEG | Inception-v3 | 2.4 | 6.8 | 1.9 | 77.0 | 1.8 | 7.1 | 2.0 | 69.8 | 2.6 | 3.6 | 2.5 | 32.1 | 0.1 | 0.4 | 0.0 | 77.0 | 0.0 | 0.2 | 0.1 | 69.8 | 0.0 | 0.2 | 0.1 | 28.0 |
| | | ResNet-101 | 1.6 | 3.5 | 2.2 | 76.6 | 1.1 | 4.2 | 2.5 | 70.5 | 2.4 | 2.8 | 2.2 | 28.1 | 0.0 | 0.1 | 0.1 | 76.6 | 0.0 | 0.1 | 0.0 | 70.5 | 0.0 | 0.0 | 0.0 | 24.0 |
| | | DenseNet-161 | 1.4 | 4.1 | 2.0 | 77.0 | 1.2 | 3.8 | 1.9 | 72.2 | 2.5 | 3.5 | 2.7 | 25.9 | 0.1 | 0.1 | 0.0 | 77.0 | 0.0 | 0.0 | 0.4 | 72.2 | 0.1 | 0.1 | 0.1 | 22.3 |
| | | AlexNet | 2.9 | 4.1 | 2.9 | 76.5 | 2.5 | 4.2 | 2.2 | 70.0 | 2.0 | 3.4 | 2.5 | 27.8 | 0.1 | 0.1 | 0.1 | 76.4 | 0.0 | 0.3 | 0.2 | 70.0 | 0.0 | 0.0 | 0.0 | 24.2 |
| | image | Inception-v3 | 99.8 | 96.4 | 3.0 | 9.3 | 99.4 | 94.7 | 3.1 | 2.9 | 46.0 | 46.8 | 2.4 | 2.5 | 93.0 | 75.1 | 0.1 | 0.5 | 98.8 | 87.0 | 0.0 | 0.0 | 13.2 | 11.7 | 0.0 | 0.0 |
| | | ResNet-101 | 99.3 | 75.0 | 2.4 | 11.0 | 98.5 | 68.5 | 2.7 | 3.5 | 34.1 | 14.7 | 2.5 | 3.7 | 90.5 | 56.4 | 0.1 | 2.0 | 96.2 | 52.4 | 0.0 | 0.1 | 9.8 | 0.2 | 0.0 | 0.0 |
| | | DenseNet-161 | 99.1 | 95.8 | 2.5 | 5.7 | 97.7 | 86.8 | 2.9 | 3.8 | 43.9 | 50.9 | 2.1 | 2.6 | 89.5 | 71.1 | 0.1 | 0.2 | 96.4 | 67.2 | 0.0 | 0.2 | 15.0 | 17.9 | 0.0 | 0.0 |
| | | AlexNet | 95.5 | 60.9 | 2.5 | 3.5 | 85.7 | 50.9 | 2.4 | 3.2 | 34.6 | 15.6 | 2.5 | 2.3 | 77.9 | 25.4 | 0.0 | 2.6 | 72.2 | 19.7 | 0.0 | 2.5 | 12.9 | 4.2 | 0.0 | 2.3 |
| Li et al. (2021) randomized | EEG | Inception-v3 | 2.6 | 2.5 | 2.5 | 1.6 | 2.6 | 2.9 | 2.6 | 2.3 | 3.0 | 2.2 | 2.7 | 1.6 | 0.0 | 0.1 | 0.1 | 1.9 | 0.1 | 0.1 | 0.2 | 2.5 | 0.2 | 0.1 | 0.2 | 1.4 |
| | | ResNet-101 | 2.6 | 2.4 | 2.8 | 2.1 | 2.8 | 2.7 | 2.9 | 2.3 | 2.6 | 2.7 | 3.0 | 1.8 | 0.0 | 0.1 | 0.1 | 1.9 | 0.0 | 0.1 | 0.1 | 2.3 | 0.1 | 0.2 | 0.1 | 1.6 |
| | | DenseNet-161 | 2.1 | 2.2 | 2.3 | 1.5 | 2.5 | 2.3 | 2.5 | 2.0 | 3.1 | 2.0 | 1.9 | 1.9 | 0.1 | 0.1 | 0.1 | 1.5 | 0.0 | 0.1 | 0.1 | 2.0 | 0.1 | 0.1 | 0.1 | 1.7 |
| | | AlexNet | 2.5 | 2.8 | 3.0 | 2.2 | 2.2 | 2.4 | 2.9 | 1.6 | 2.6 | 2.8 | 3.1 | 2.4 | 0.2 | 0.1 | 0.2 | 2.2 | 0.3 | 0.1 | 0.1 | 1.6 | 0.1 | 0.1 | 0.0 | 1.9 |
| | image | Inception-v3 | 99.8 | 91.1 | 3.4 | 9.4 | 99.4 | 84.7 | 3.0 | 4.9 | 46.0 | 57.1 | 2.1 | 2.2 | 93.0 | 64.0 | 0.1 | 73.3 | 98.8 | 73.3 | 0.0 | 0.0 | 13.2 | 20.4 | 0.0 | 0.0 |
| | | ResNet-101 | 99.3 | 69.8 | 2.6 | 11.2 | 98.5 | 53.1 | 2.6 | 4.0 | 34.1 | 9.4 | 3.1 | 2.2 | 90.5 | 42.0 | 0.0 | 28.5 | 96.2 | 28.5 | 0.1 | 0.1 | 9.8 | 3.5 | 0.0 | 0.0 |
| | | DenseNet-161 | 99.1 | 65.6 | 2.1 | 5.1 | 97.7 | 36.0 | 2.5 | 3.0 | 43.9 | 36.1 | 2.3 | 3.0 | 89.5 | 30.5 | 0.1 | 11.7 | 96.4 | 11.7 | 0.0 | 0.0 | 15.0 | 13.6 | 0.0 | 0.0 |
| | | AlexNet | 95.5 | 3.6 | 2.4 | 3.5 | 85.7 | 3.6 | 2.5 | 3.5 | 34.6 | 2.5 | 2.5 | 2.5 | 77.9 | 0.0 | 0.0 | 0.0 | 72.2 | 0.0 | 0.0 | 2.6 | 12.9 | 0.0 | 0.0 | 2.5 |

Table 3: Per-sample triplet loss on the training set, after joint training, both with and without 'pretraining,' and after separate training, averaged over samples and splits.

| | | joint | | | | | | separate | | |
| | | pretraining | | | no pretraining | | | | | |
| | | Spampinato et al. (2017) | Li et al. (2021) block | Li et al. (2021) randomized | Spampinato et al. (2017) | Li et al. (2021) block | Li et al. (2021) randomized | Spampinato et al. (2017) | Li et al. (2021) block | Li et al. (2021) randomized |
| | | i | ii | iii | iv | v | vi | vii | viii | ix |
| Inception v3 | i | 1.303 | 1.098 | 0.932 | 1.074 | 0.938 | 0.919 | 1.150 | 1.027 | 1.104 |
| ResNet-101 | ii | 1.155 | 0.925 | 1.019 | 0.963 | 1.006 | 1.080 | 0.885 | 0.936 | 0.884 |
| DenseNet-161 | iii | 1.027 | 1.080 | 1.047 | 0.988 | 0.919 | 0.929 | 1.007 | 0.982 | 1.029 |
| AlexNet | iv | 1.417 | 1.142 | 0.981 | 1.136 | 1.055 | 1.034 | 0.925 | 0.956 | 0.935 |

Table 4: Per-sample MSE loss of the individual encoders on the training set, after separate training, averaged over samples and splits.

| | | EEG | | | image | | |
| | | Spampinato et al. (2017) | Li et al. (2021) block | Li et al. (2021) randomized | Spampinato et al. (2017) | Li et al. (2021) block | Li et al. (2021) randomized |
| | | i | ii | iii | iv | v | vi |
| Inception v3 | i | 0.001 | 0.001 | 0.001 | 0.002 | 0.010 | 0.010 |
| ResNet-101 | ii | 0.001 | 0.001 | 0.001 | 0.001 | 0.002 | 0.002 |
| DenseNet-161 | iii | 0.001 | 0.001 | 0.001 | 0.001 | 0.002 | 0.002 |
| AlexNet | iv | 0.001 | 0.001 | 0.001 | 0.001 | 0.001 | 0.001 |

Recent progress in computer vision and machine learning has been gauged largely by improvement in performance metrics of methods on datasets and results that 'look good.' Palazzo et al. (2018; 2020a;b; 2021) largely base their claims on similar criteria. While synthetic engineering methods can be evaluated by the utility of artifacts, analytic scientific claims about the underlying physical world can only be evaluated by finding hypotheses that resist falsification. We have falsified all of the claims of Palazzo et al. (2018; 2020a;b; 2021).

AUTHOR CONTRIBUTIONS

Removed for blind review.

ACKNOWLEDGMENTS

Removed for blind review.

ETHICS STATEMENT

This work debunks nearly one hundred published papers whose results are based on the same confound: a correlation between stimulus class and temporal drift. This confound has been found in eighteen available EEG datasets. Just as with an inconsistent set of axioms one can prove anything, a confounded dataset can be used to support any claim, even ones that are false or absurd. That is what many recent publications based on this confound do: things like generating high fidelity renderings of images, or even 3D CAD models of objects, from EEG recordings.

A research community, knowingly or unknowingly, has discovered that one can use confounded datasets to churn out a plethora of flawed results without reviewers noticing. They have also discovered that one can collect new confounded datasets to churn out even more flawed results without reviewers noticing. The temptation to do this is so strong that the community continues to do so four years after details of the confound were published.

It is conceivable that the flaws in these datasets may be a driving factor behind their frequent reuse. When a dataset is severely confounded, it becomes relatively easy to achieve an extremely high accuracy, which can in turn be used to support sensational claims, and ultimately directs further attention to the dataset. In business, this phenomenon is referred to as "the bad money drives out the good money."

More prominent exposure of these flawed methods and consequent false results will allow resources wasted on continued use of these confounded datasets and flawed methods to be reallocated. The debunked work also causes direct ongoing harm:

- grant proposals can be rejected due to preliminary results not being competitive with results demonstrating falsely-inflated performance based on confounded data or faulty methods;
- manuscripts can be rejected for the same reason;
- grants can be awarded based on false pretenses
- manuscripts can be accepted for the same reason;
- degrees can be awarded for the same reason;

- resources can be wasted attempting to replicate the debunked results;
- resources can be wasted having people read and review flawed papers, and learn flawed methods; and
- because the debunked work relates to brain-computer interfaces—whose primary application is helping people with disabilities (*e.g.*, paralysis) interact with the world—the harm caused is not merely scientific but also medical, with disproportionate impact on people with disabilities.

## REPRODUCIBILITY STATEMENT

The raw data that produced these results is available at `http://dx.doi.org/10.21227/x2gf-5324`. Our code, which will be released upon publication, is built on top of the code in `http://dx.doi.org/10.21227/x2gf-5324`.

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

## A  APPENDIX

## B  METHOD

We report all results for three sets of EEG data: the data reported by Spampinato et al. (2017) as used by Palazzo et al. (2018; 2020a;b; 2021) ('Spampinato et al. (2017)'), the data reported by Li et al. (2021) for subject 6 first image block run ('Li et al. (2021) block'), and the data reported by Li et al. (2021) for subject 6 image rapid-event run ('Li et al. (2021) randomized'). We report all results for four off-the-shelf image classifiers taken as image encoders: Inception v3 (Szegedy et al., 2016), ResNet-101 (He et al., 2016), DenseNet-161 (Huang et al., 2017), and AlexNet (Krizhevsky et al., 2012). We report all results both before joint training ('before'), where the EEG encoder is randomly initialized, after joint training ('after joint'), and after separate training ('after separate'). For 'before' and 'after separate' the image encoder is initialized with off-the-shelf weights trained on the ILSVRC 2012 training set (Russakovsky et al., 2014). For 'joint', we report results both where the image encoder is initialized with off-the-shelf weights trained on the ILSVRC 2012 training set (Russakovsky et al., 2014) ('pretraining') and where the image encoder is randomly initialized ('no pretraining'). The EEG and image encoders both output a 1000 element encoding. During joint training, a triplet loss is used to drive the EEG and image encodings to have a high dot product when the EEG signal is associated with the stimulus image that elicited the associated EEG signal and a low dot product when the EEG signal is associated with a different image, irrespective of class. We do not postpend a classifier (ReLU, 1000→40 FC, softmax) to the encoders, either during training or after training. All analyses are done directly on the unmodified encodings produced by the encoders. For both the EEG encodings ('EEG') and the image encodings ('image'), we analyze the unmodified encodings ('original') and encodings reconstructed by principal component analysis ('PCA'). For PCA, we analyze two reconstructions: one with only the top 40 principal components ('top 40') and one with only the bottom 960 principal components ('bottom 960'). We classify original and reconstructed EEG and image encodings, before and after joint and separate training, in two ways. In the first, we simply select the label of the maximal element among all 1000 elements ('1000 classes'). In the second, we select the label of the maximal element among only the 40 classes used as stimuli ('40 classes'). In both cases, correct labels are considered positives. In the former, negatives can result from incorrect labels, both within and outside the subset of 40 ILSVRC classes used as stimuli. In the latter, negatives can result only from incorrect labels within the subset of 40 ILSVRC classes used as stimuli.

Table 5: The top 40 principal components explain a large portion of the image-encoding variance (≥63.9%) before training (column v). This together with Table 2 implies that the pretrained image encoders produce encodings that contain (primarily) class information on all data. This supports claim A and calls claim I into question.

| | | | EEG before | EEG after joint pretraining | EEG after joint no pretraining | EEG after separate | image before | image after joint pretraining | image after joint no pretraining | image after separate |
|---|---|---|---|---|---|---|---|---|---|---|
| | | | i | ii | iii | iv | v | vi | vii | viii |
| Spampinato et al. (2017) | Inception v3 | i | 44.7 | 37.2 | 15.3 | 99.5 | 64.1 | 94.9 | 98.9 | 100.0 |
| | ResNet-101 | ii | 46.5 | 36.2 | 17.7 | 99.5 | 82.0 | 98.9 | 98.2 | 96.8 |
| | DenseNet-161 | iii | 48.2 | 25.6 | 22.7 | 100.0 | 75.4 | 97.4 | 90.2 | 100.0 |
| | AlexNet | iv | 46.3 | 21.7 | 18.5 | 99.5 | 82.3 | nan | 100.0 | 99.7 |
| Li et al. (2021) block | Inception v3 | v | 83.3 | 75.9 | 69.4 | 42.3 | 63.9 | 57.4 | 98.3 | 99.9 |
| | ResNet-101 | vi | 85.4 | 79.8 | 72.8 | 41.9 | 81.9 | 99.5 | 99.0 | 99.7 |
| | DenseNet-161 | vii | 84.0 | 79.3 | 76.3 | 42.9 | 75.3 | 69.4 | 94.3 | 99.5 |
| | AlexNet | viii | 84.4 | 78.9 | 73.2 | 40.0 | 82.2 | 78.9 | 99.9 | 98.9 |
| Li et al. (2021) randomized | Inception v3 | ix | 62.1 | 40.5 | 38.3 | 19.3 | 63.9 | 60.2 | 98.1 | 99.9 |
| | ResNet-101 | x | 61.9 | 52.2 | 43.6 | 19.7 | 81.9 | 99.1 | 98.8 | 99.8 |
| | DenseNet-161 | xi | 63.5 | 48.5 | 46.5 | 19.6 | 75.3 | 70.8 | 93.9 | 99.7 |
| | AlexNet | xii | 63.6 | 49.5 | 41.3 | 19.2 | 82.2 | 86.9 | 99.9 | 98.8 |

Table 6: After joint training with pretraining on the 'Spampinato et al. (2017)' data, the top 40 principal components explain almost all (≥94.9%) of the variance in the image encodings (column vi, for rows i–iv). This implies that the image encoders jointly trained on the 'Spampinato et al. (2017)' data produce encodings with little more than class information.

| | | | EEG before | EEG after joint pretraining | EEG after joint no pretraining | EEG after separate | image before | image after joint pretraining | image after joint no pretraining | image after separate |
|---|---|---|---|---|---|---|---|---|---|---|
| | | | i | ii | iii | iv | v | vi | vii | viii |
| Spampinato et al. (2017) | Inception v3 | i | 44.7 | 37.2 | 15.3 | 99.5 | 64.1 | 94.9 | 98.9 | 100.0 |
| | ResNet-101 | ii | 46.5 | 36.2 | 17.7 | 99.5 | 82.0 | 98.9 | 98.2 | 96.8 |
| | DenseNet-161 | iii | 48.2 | 25.6 | 22.7 | 100.0 | 75.4 | 97.4 | 90.2 | 100.0 |
| | AlexNet | iv | 46.3 | 21.7 | 18.5 | 99.5 | 82.3 | nan | 100.0 | 99.7 |
| Li et al. (2021) block | Inception v3 | v | 83.3 | 75.9 | 69.4 | 42.3 | 63.9 | 57.4 | 98.3 | 99.9 |
| | ResNet-101 | vi | 85.4 | 79.8 | 72.8 | 41.9 | 81.9 | 99.5 | 99.0 | 99.7 |
| | DenseNet-161 | vii | 84.0 | 79.3 | 76.3 | 42.9 | 75.3 | 69.4 | 94.3 | 99.5 |
| | AlexNet | viii | 84.4 | 78.9 | 73.2 | 40.0 | 82.2 | 78.9 | 99.9 | 98.9 |
| Li et al. (2021) randomized | Inception v3 | ix | 62.1 | 40.5 | 38.3 | 19.3 | 63.9 | 60.2 | 98.1 | 99.9 |
| | ResNet-101 | x | 61.9 | 52.2 | 43.6 | 19.7 | 81.9 | 99.1 | 98.8 | 99.8 |
| | DenseNet-161 | xi | 63.5 | 48.5 | 46.5 | 19.6 | 75.3 | 70.8 | 93.9 | 99.7 |
| | AlexNet | xii | 63.6 | 49.5 | 41.3 | 19.2 | 82.2 | 86.9 | 99.9 | 98.8 |

Table 7: After separate training, the top 40 principal components explain almost all of the variance (≥96.8%) in the image encodings (column viii). This implies that the separately trained image encoders produce encodings with little more than class information on all data.

| | | | EEG before | EEG after joint pretraining | EEG after joint no pretraining | EEG after separate | image before | image after joint pretraining | image after joint no pretraining | image after separate |
|---|---|---|---|---|---|---|---|---|---|---|
| | | | i | ii | iii | iv | v | vi | vii | viii |
| Spampinato et al. (2017) | Inception v3 | i | 44.7 | 37.2 | 15.3 | 99.5 | 64.1 | 94.9 | 98.9 | 100.0 |
| | ResNet-101 | ii | 46.5 | 36.2 | 17.7 | 99.5 | 82.0 | 98.9 | 98.2 | 96.8 |
| | DenseNet-161 | iii | 48.2 | 25.6 | 22.7 | 100.0 | 75.4 | 97.4 | 90.2 | 100.0 |
| | AlexNet | iv | 46.3 | 21.7 | 18.5 | 99.5 | 82.3 | nan | 100.0 | 99.7 |
| Li et al. (2021) block | Inception v3 | v | 83.3 | 75.9 | 69.4 | 42.3 | 63.9 | 57.4 | 98.3 | 99.9 |
| | ResNet-101 | vi | 85.4 | 79.8 | 72.8 | 41.9 | 81.9 | 99.5 | 99.0 | 99.7 |
| | DenseNet-161 | vii | 84.0 | 79.3 | 76.3 | 42.9 | 75.3 | 69.4 | 94.3 | 99.5 |
| | AlexNet | viii | 84.4 | 78.9 | 73.2 | 40.0 | 82.2 | 78.9 | 99.9 | 98.9 |
| Li et al. (2021) randomized | Inception v3 | ix | 62.1 | 40.5 | 38.3 | 19.3 | 63.9 | 60.2 | 98.1 | 99.9 |
| | ResNet-101 | x | 61.9 | 52.2 | 43.6 | 19.7 | 81.9 | 99.1 | 98.8 | 99.8 |
| | DenseNet-161 | xi | 63.5 | 48.5 | 46.5 | 19.6 | 75.3 | 70.8 | 93.9 | 99.7 |
| | AlexNet | xii | 63.6 | 49.5 | 41.3 | 19.2 | 82.2 | 86.9 | 99.9 | 98.8 |

Table 8: After separate training, the top 40 principal components explain almost all of the variance ($\geq$96.8%) in both the EEG and image encodings on the 'Spampinato et al. (2017)' data (columns iv and viii, for rows i–iv). This implies that the separately trained EEG and image encoders produce encodings with little more than class information on the 'Spampinato et al. (2017)' data.

| | | | EEG | | | | image | | | |
| | | | before | after joint | | after separate | before | after joint | | after separate |
| | | | | pretraining | no pretraining | | | pretraining | no pretraining | |
| | | | i | ii | iii | iv | v | vi | vii | viii |
|---|---|---|---|---|---|---|---|---|---|---|
| Spampinato et al. (2017) | Inception v3 | i | 44.7 | 37.2 | 15.3 | 99.5 | 64.1 | 94.9 | 98.9 | 100.0 |
| | ResNet-101 | ii | 46.5 | 36.2 | 17.7 | 99.5 | 82.0 | 98.9 | 98.2 | 96.8 |
| | DenseNet-161 | iii | 48.2 | 25.6 | 22.7 | 100.0 | 75.4 | 97.4 | 90.2 | 100.0 |
| | AlexNet | iv | 46.3 | 21.7 | 18.5 | 99.5 | 82.3 | nan | 100.0 | 99.7 |
| Li et al. (2021) block | Inception v3 | v | 83.3 | 75.9 | 69.4 | 42.3 | 63.9 | 57.4 | 98.3 | 99.9 |
| | ResNet-101 | vi | 85.4 | 79.8 | 72.8 | 41.9 | 81.9 | 99.5 | 99.0 | 99.7 |
| | DenseNet-161 | vii | 84.0 | 79.3 | 76.3 | 42.9 | 75.3 | 69.4 | 94.3 | 99.5 |
| | AlexNet | viii | 84.4 | 78.9 | 73.2 | 40.0 | 82.2 | 78.9 | 99.9 | 98.9 |
| Li et al. (2021) randomized | Inception v3 | ix | 62.1 | 40.5 | 38.3 | 19.3 | 63.9 | 60.2 | 98.1 | 99.9 |
| | ResNet-101 | x | 61.9 | 52.2 | 43.6 | 19.7 | 81.9 | 99.1 | 98.8 | 99.8 |
| | DenseNet-161 | xi | 63.5 | 48.5 | 46.5 | 19.6 | 75.3 | 70.8 | 93.9 | 99.7 |
| | AlexNet | xii | 63.6 | 49.5 | 41.3 | 19.2 | 82.2 | 86.9 | 99.9 | 98.8 |

Table 9: Image classification accuracy is near perfect with all components with 40 classes (≥95.5%; column i, for rows v–viii, xiii–xvi, and xxi–xxiv) before training. This holds both for the 'Spampinato et al. (2017)' data, the 'Li et al. (2021) block' data, and the 'Li et al. (2021) randomized' data. This, together with Tables 10 and 11, demonstrates that they are giving class information implicitly to their training by use of image encoders pretrained on ImageNet, supports claim A, and calls claim I into question.

| Dataset | Mod. | Model | # | i (40: orig. before) | ii (joint pretr.) | iii (joint no-pretr.) | iv (separate) | v (40: top40 before) | vi (joint pretr.) | vii (no-pretr.) | viii (separate) | ix (40: bot960 before) | x (joint pretr.) | xi (no-pretr.) | xii (separate) | xiii (1000: orig. before) | xiv (joint pretr.) | xv (no-pretr.) | xvi (separate) | xvii (1000: top40 before) | xviii (joint pretr.) | xix (no-pretr.) | xx (separate) | xxi (1000: bot960 before) | xxii (joint pretr.) | xxiii (no-pretr.) | xxiv (separate) |
|---|---|---|---|---|---|---|---|---|---|---|---|---|---|---|---|---|---|---|---|---|---|---|---|---|---|---|---|
| Spampinato et al. (2017) | EEG | Inception-v3 | i | 2.6 | 3.0 | 2.5 | 55.5 | 2.2 | 2.6 | 2.5 | 55.5 | 2.5 | 3.1 | 2.5 | 2.3 | 0.1 | 0.1 | 0.1 | 55.5 | 0.1 | 0.1 | 0.1 | 55.5 | 0.1 | 0.1 | 0.1 | 2.3 |
| | | ResNet-101 | ii | 2.3 | 3.0 | 2.6 | 54.7 | 2.8 | 2.4 | 2.4 | 54.7 | 2.4 | 3.1 | 2.5 | 2.3 | 0.0 | 0.1 | 0.1 | 54.7 | 0.0 | 0.0 | 0.1 | 54.7 | 0.1 | 0.1 | 0.1 | 2.3 |
| | | DenseNet-161 | iii | 2.5 | 3.1 | 2.4 | 57.4 | 2.4 | 2.6 | 2.3 | 57.4 | 2.5 | 3.2 | 2.4 | 2.5 | 0.1 | 0.1 | 0.1 | 57.4 | 0.0 | 0.1 | 0.0 | 57.4 | 0.1 | 0.1 | 0.1 | 2.5 |
| | | AlexNet | iv | 2.3 | 2.5 | 2.4 | 57.3 | 2.4 | 2.6 | 2.6 | 57.3 | 2.4 | 2.5 | 2.6 | 2.5 | 0.1 | 0.1 | 0.1 | 57.3 | 0.0 | 0.1 | 0.1 | 57.3 | 0.1 | 0.1 | 0.1 | 2.5 |
| | image | Inception-v3 | v | **99.7** | 43.2 | 2.7 | 3.4 | 99.4 | 13.5 | 2.9 | 2.2 | 45.5 | 29.4 | 2.7 | 2.5 | 92.8 | 16.5 | 0.0 | 0.0 | 98.9 | 4.0 | 0.0 | 0.0 | 12.8 | 9.3 | 0.0 | 0.0 |
| | | ResNet-101 | vi | **99.4** | 24.2 | 2.4 | 82.5 | 98.3 | 7.3 | 2.9 | 70.5 | 33.0 | 9.5 | 2.1 | 39.4 | 90.5 | 2.6 | 0.0 | 82.5 | 95.6 | 0.3 | 0.3 | 70.5 | 9.5 | 0.7 | 0.0 | 39.2 |
| | | DenseNet-161 | vii | **99.2** | 7.2 | 2.4 | 23.3 | 97.9 | 4.4 | 2.3 | 3.5 | 42.7 | 3.8 | 2.6 | 2.6 | 89.4 | 0.1 | 0.0 | 12.1 | 95.8 | 0.0 | 0.0 | 0.1 | 14.2 | 0.0 | 0.1 | 0.0 |
| | | AlexNet | viii | **95.8** | 2.6 | 2.7 | 18.7 | 85.8 | 2.6 | 2.7 | 19.1 | 34.1 | 2.6 | 2.4 | 2.4 | 78.4 | 0.0 | 0.3 | 18.7 | 72.5 | 0.0 | 0.3 | 19.1 | 12.5 | 0.0 | 0.0 | 2.4 |
| Li et al. (2021) Block | EEG | Inception-v3 | ix | 2.4 | 6.8 | 1.9 | 77.0 | 1.8 | 7.1 | 2.0 | 69.8 | 2.6 | 3.6 | 2.5 | 32.1 | 0.1 | 0.4 | 0.0 | 77.0 | 0.0 | 0.2 | 0.1 | 69.8 | 0.0 | 0.2 | 0.1 | 28.0 |
| | | ResNet-101 | x | 1.6 | 3.5 | 2.2 | 76.6 | 1.1 | 4.2 | 2.5 | 70.5 | 2.4 | 2.8 | 2.2 | 28.1 | 0.0 | 0.1 | 0.1 | 76.6 | 0.0 | 0.0 | 0.1 | 70.5 | 0.0 | 0.0 | 0.1 | 24.0 |
| | | DenseNet-161 | xi | 1.4 | 4.1 | 2.0 | 77.0 | 1.2 | 3.8 | 1.9 | 72.2 | 2.5 | 3.5 | 2.7 | 25.9 | 0.1 | 0.1 | 0.0 | 77.0 | 0.0 | 0.0 | 0.0 | 72.2 | 0.1 | 0.1 | 0.1 | 22.3 |
| | | AlexNet | xii | 2.9 | 4.1 | 2.9 | 76.5 | 2.5 | 4.2 | 2.2 | 70.0 | 2.0 | 3.4 | 2.7 | 27.8 | 0.0 | 0.1 | 0.1 | 76.4 | 0.0 | 0.3 | 0.2 | 70.0 | 0.0 | 0.1 | 0.1 | 24.2 |
| | image | Inception-v3 | xiii | **99.8** | 96.4 | 3.0 | 9.3 | 99.4 | 94.7 | 3.1 | 2.9 | 46.0 | 46.8 | 2.4 | 2.5 | 93.0 | 75.1 | 0.1 | 0.5 | 98.8 | 87.0 | 0.0 | 0.0 | 13.2 | 11.7 | 0.0 | 0.0 |
| | | ResNet-101 | xiv | **99.3** | 75.0 | 2.4 | 11.0 | 98.5 | 68.5 | 2.7 | 3.5 | 34.1 | 14.7 | 2.5 | 3.7 | 90.5 | 56.4 | 0.0 | 2.0 | 96.2 | 52.4 | 0.1 | 0.1 | 9.8 | 0.2 | 0.0 | 0.0 |
| | | DenseNet-161 | xv | **99.1** | 95.8 | 2.5 | 5.7 | 97.7 | 86.8 | 2.9 | 3.8 | 43.9 | 50.9 | 2.1 | 2.6 | 89.5 | 71.1 | 0.1 | 0.2 | 96.4 | 67.2 | 0.4 | 0.2 | 15.0 | 17.9 | 0.0 | 0.0 |
| | | AlexNet | xvi | **95.5** | 60.9 | 2.5 | 3.5 | 85.7 | 50.9 | 2.4 | 3.2 | 34.6 | 15.6 | 2.5 | 2.3 | 77.9 | 25.4 | 0.0 | 2.6 | 72.2 | 19.7 | 0.0 | 2.5 | 12.9 | 4.2 | 0.0 | 2.3 |
| Li et al. (2021) randomized | EEG | Inception-v3 | xvii | 2.6 | 2.5 | 2.5 | 1.6 | 2.6 | 2.9 | 2.6 | 2.5 | 3.0 | 2.2 | 2.7 | 1.6 | 0.1 | 0.1 | 0.1 | 1.5 | 0.1 | 0.1 | 0.0 | 2.5 | 0.2 | 0.1 | 0.2 | 1.4 |
| | | ResNet-101 | xviii | 2.6 | 2.4 | 2.8 | 2.1 | 2.8 | 2.7 | 2.9 | 2.3 | 2.6 | 2.7 | 3.0 | 1.8 | 0.0 | 0.1 | 0.1 | 1.9 | 0.0 | 0.1 | 0.2 | 2.3 | 0.1 | 0.2 | 0.1 | 1.6 |
| | | DenseNet-161 | xix | 2.1 | 2.2 | 2.3 | 1.5 | 2.5 | 2.3 | 2.5 | 2.0 | 3.1 | 2.0 | 1.9 | 1.9 | 0.1 | 0.1 | 0.1 | 1.5 | 0.0 | 0.0 | 0.1 | 2.0 | 0.1 | 0.1 | 0.1 | 1.7 |
| | | AlexNet | xx | 2.5 | 2.8 | 3.0 | 2.2 | 2.2 | 2.4 | 2.9 | 1.6 | 2.6 | 2.8 | 3.1 | 2.4 | 0.2 | 0.1 | 0.2 | 2.2 | 0.3 | 0.0 | 0.1 | 1.6 | 0.1 | 0.1 | 0.1 | 1.9 |
| | image | Inception-v3 | xxi | **99.8** | 91.1 | 3.4 | 9.4 | 99.4 | 84.7 | 3.0 | 2.9 | 46.0 | 57.1 | 2.1 | 2.5 | 93.0 | 64.0 | 0.1 | 0.6 | 98.8 | 73.3 | 0.0 | 0.0 | 13.2 | 20.4 | 0.0 | 0.0 |
| | | ResNet-101 | xxii | **99.3** | 69.8 | 2.9 | 11.2 | 98.5 | 53.1 | 2.6 | 4.0 | 34.1 | 9.4 | 3.1 | 2.2 | 90.5 | 42.0 | 0.0 | 2.0 | 96.2 | 28.5 | 0.0 | 0.1 | 9.8 | 3.5 | 0.0 | 0.0 |
| | | DenseNet-161 | xxiii | **99.1** | 65.6 | 2.1 | 5.1 | 97.7 | 36.0 | 2.5 | 3.0 | 43.9 | 36.1 | 2.3 | 3.0 | 89.5 | 30.5 | 0.1 | 0.0 | 96.4 | 11.7 | 0.0 | 0.0 | 15.0 | 13.6 | 0.0 | 0.0 |
| | | AlexNet | xxiv | **95.5** | 3.6 | 2.4 | 3.5 | 85.7 | 3.6 | 2.5 | 3.5 | 34.6 | 2.5 | 2.5 | 2.5 | 77.9 | 0.0 | 0.0 | 2.6 | 72.2 | 0.0 | 0.0 | 2.6 | 12.9 | 0.0 | 0.0 | 2.5 |

Table 10: Image classification accuracy is very high with all components with 1000 classes (≥77.9%; column xiii, for rows v–viii, xiii–xvi, and xxi–xxiv) before training. This holds both for the 'Spampinato et al. (2017)' data, the 'Li et al. (2021) block' data, and the 'Li et al. (2021) randomized' data. This, together with Tables 9 and 11, demonstrates that they are giving class information implicitly to their training by use of image encoders pretrained on ImageNet, supports claim A, and calls claim I into question.

| | | | | original | | | | 40 classes / top 40 | | | | bottom 960 | | | | original | | | | 1000 classes / top 40 | | | | bottom 960 | | | |
|---|---|---|---|---|---|---|---|---|---|---|---|---|---|---|---|---|---|---|---|---|---|---|---|---|---|---|---|
| | | | | before | after joint pretr. | no pretr. | after sep. | before | after joint pretr. | no pretr. | after sep. | before | after joint pretr. | no pretr. | after sep. | before | after joint pretr. | no pretr. | after sep. | before | after joint pretr. | no pretr. | after sep. | before | after joint pretr. | no pretr. | after sep. |
| | | | | i | ii | iii | iv | v | vi | vii | viii | ix | x | xi | xii | xiii | xiv | xv | xvi | xvii | xviii | xix | xx | xxi | xxii | xxiii | xxiv |
| Spampinato et al. (2017) | EEG | Inception v3 | i | 2.6 | 3.0 | 2.5 | 55.5 | 2.2 | 2.6 | 2.5 | 55.5 | 2.5 | 3.1 | 2.5 | 2.3 | 0.1 | 0.1 | 0.1 | 55.5 | 0.1 | 0.1 | 0.1 | 55.5 | 0.1 | 0.1 | 0.1 | 2.3 |
| | | ResNet-101 | ii | 2.3 | 3.0 | 2.6 | 54.7 | 2.8 | 2.4 | 2.4 | 54.7 | 2.4 | 3.1 | 2.5 | 2.3 | 0.0 | 0.1 | 0.1 | 54.7 | 0.0 | 0.0 | 0.1 | 54.7 | 0.1 | 0.1 | 0.1 | 2.3 |
| | | DenseNet-161 | iii | 2.5 | 3.1 | 2.4 | 57.4 | 2.4 | 2.6 | 2.3 | 57.4 | 2.5 | 3.2 | 2.4 | 2.5 | 0.1 | 0.1 | 0.1 | 57.4 | 0.0 | 0.1 | 0.0 | 57.4 | 0.1 | 0.1 | 0.1 | 2.5 |
| | | AlexNet | iv | 2.3 | 2.5 | 2.4 | 57.3 | 2.4 | 2.6 | 2.6 | 57.3 | 2.4 | 2.5 | 2.6 | 2.5 | 0.1 | 0.1 | 0.1 | 57.3 | 0.0 | 0.1 | 0.1 | 57.3 | 0.1 | 0.1 | 0.1 | 2.5 |
| | image | Inception v3 | v | 99.7 | 43.2 | 2.7 | 3.4 | 99.4 | 13.5 | 2.9 | 2.2 | 45.5 | 29.4 | 2.7 | 2.5 | 92.8 | 16.5 | 0.1 | 0.0 | 98.9 | 4.0 | 0.0 | 0.0 | 12.8 | 9.3 | 0.0 | 0.0 |
| | | ResNet-101 | vi | 99.4 | 24.2 | 2.4 | 82.5 | 98.3 | 7.3 | 2.9 | 70.5 | 33.0 | 9.5 | 2.1 | 39.4 | 90.5 | 2.6 | 0.0 | 82.5 | 95.6 | 0.3 | 0.0 | 70.5 | 9.5 | 0.7 | 0.0 | 39.2 |
| | | DenseNet-161 | vii | 99.2 | 7.2 | 2.4 | 23.3 | 97.9 | 4.4 | 2.3 | 3.5 | 42.7 | 3.8 | 2.6 | 2.6 | 89.4 | 0.1 | 0.0 | 12.1 | 95.8 | 0.0 | 0.0 | 0.1 | 14.2 | 0.0 | 0.1 | 0.0 |
| | | AlexNet | viii | 95.8 | 2.6 | 2.7 | 18.7 | 85.8 | 2.6 | 2.7 | 19.1 | 34.1 | 2.6 | 2.4 | 2.4 | 78.4 | 0.0 | 0.3 | 18.7 | 72.5 | 0.0 | 0.3 | 19.1 | 12.5 | 0.0 | 0.0 | 2.4 |
| Li et al. (2021) block | EEG | Inception v3 | ix | 2.4 | 6.8 | 1.9 | 77.0 | 1.8 | 7.1 | 2.0 | 69.8 | 2.6 | 3.6 | 2.5 | 32.1 | 0.1 | 0.4 | 0.0 | 77.0 | 0.0 | 0.2 | 0.1 | 69.8 | 0.0 | 0.2 | 0.1 | 28.0 |
| | | ResNet-101 | x | 1.6 | 3.5 | 2.2 | 76.6 | 1.1 | 4.2 | 2.5 | 70.5 | 2.4 | 2.8 | 2.2 | 28.1 | 0.0 | 0.1 | 0.1 | 76.5 | 0.0 | 0.1 | 0.1 | 70.5 | 0.0 | 0.0 | 0.1 | 24.0 |
| | | DenseNet-161 | xi | 1.4 | 4.1 | 2.0 | 77.0 | 1.2 | 3.8 | 1.9 | 72.2 | 2.5 | 3.5 | 2.7 | 25.9 | 0.1 | 0.1 | 0.0 | 77.0 | 0.0 | 0.0 | 0.0 | 72.2 | 0.1 | 0.1 | 0.1 | 22.3 |
| | | AlexNet | xii | 2.9 | 4.1 | 2.9 | 76.5 | 2.5 | 4.2 | 2.2 | 70.0 | 2.0 | 3.4 | 2.9 | 27.8 | 0.0 | 0.1 | 0.1 | 76.4 | 0.0 | 0.3 | 0.2 | 70.0 | 0.0 | 0.1 | 0.2 | 24.2 |
| | image | Inception v3 | xiii | 99.8 | 96.4 | 3.0 | 9.3 | 99.4 | 94.7 | 3.1 | 2.9 | 46.0 | 46.8 | 2.4 | 2.9 | 93.0 | 75.1 | 0.1 | 0.5 | 98.8 | 87.0 | 0.0 | 0.0 | 13.2 | 11.7 | 0.0 | 0.0 |
| | | ResNet-101 | xiv | 99.3 | 75.0 | 2.4 | 11.0 | 98.5 | 68.5 | 2.7 | 3.5 | 34.1 | 14.7 | 2.5 | 3.7 | 90.5 | 56.4 | 0.0 | 2.0 | 96.2 | 52.4 | 0.0 | 0.1 | 9.8 | 0.2 | 0.0 | 0.0 |
| | | DenseNet-161 | xv | 99.1 | 95.8 | 2.5 | 5.7 | 97.7 | 86.8 | 2.9 | 3.8 | 43.9 | 50.9 | 2.1 | 2.6 | 89.5 | 71.1 | 0.1 | 0.2 | 96.4 | 67.2 | 0.4 | 0.2 | 15.0 | 17.9 | 0.0 | 0.0 |
| | | AlexNet | xvi | 95.5 | 60.9 | 2.5 | 3.5 | 85.7 | 50.9 | 2.4 | 3.2 | 34.6 | 15.6 | 2.5 | 2.3 | 77.9 | 25.4 | 0.0 | 2.6 | 72.2 | 19.7 | 0.0 | 2.5 | 12.9 | 4.2 | 0.0 | 2.3 |
| Li et al. (2021) randomized | EEG | Inception v3 | xvii | 2.6 | 2.5 | 2.5 | 1.6 | 2.6 | 2.9 | 2.6 | 2.5 | 3.0 | 2.2 | 2.7 | 1.6 | 0.1 | 0.1 | 0.1 | 1.5 | 0.1 | 0.1 | 0.0 | 2.5 | 0.2 | 0.1 | 0.2 | 1.4 |
| | | ResNet-101 | xviii | 2.1 | 2.4 | 2.8 | 2.1 | 2.8 | 2.7 | 2.9 | 2.3 | 2.6 | 2.7 | 3.0 | 1.8 | 0.0 | 0.1 | 0.1 | 1.9 | 0.0 | 0.1 | 0.2 | 2.3 | 0.1 | 0.2 | 0.1 | 1.6 |
| | | DenseNet-161 | xix | 2.5 | 2.2 | 2.3 | 1.5 | 2.5 | 2.3 | 2.5 | 2.0 | 3.1 | 2.0 | 1.9 | 1.9 | 0.1 | 0.1 | 0.1 | 1.5 | 0.0 | 0.0 | 0.1 | 2.0 | 0.1 | 0.1 | 0.1 | 1.7 |
| | | AlexNet | xx | | 2.8 | 3.0 | 2.2 | 2.2 | 2.4 | 2.9 | 1.6 | 2.6 | 2.8 | 3.1 | 2.4 | 0.2 | 0.1 | 0.2 | 2.2 | 0.3 | 0.0 | 0.1 | 1.6 | 0.1 | 0.1 | 0.1 | 1.9 |
| | image | Inception v3 | xxi | 99.8 | 91.1 | 3.4 | 9.4 | 99.4 | 84.7 | 3.0 | 2.9 | 46.0 | 57.1 | 2.1 | 2.5 | 93.0 | 64.0 | 0.1 | 0.6 | 98.8 | 73.3 | 0.0 | 0.0 | 13.2 | 20.4 | 0.0 | 0.0 |
| | | ResNet-101 | xxii | 99.3 | 69.8 | 2.9 | 11.2 | 98.5 | 53.1 | 2.6 | 4.0 | 34.1 | 9.4 | 3.1 | 2.2 | 90.5 | 42.0 | 0.0 | 2.0 | 96.2 | 28.5 | 0.0 | 0.1 | 9.8 | 3.5 | 0.0 | 0.0 |
| | | DenseNet-161 | xxiii | 99.1 | 65.6 | 2.1 | 5.1 | 97.7 | 36.0 | 2.5 | 3.0 | 43.9 | 36.1 | 2.3 | 3.0 | 89.5 | 30.5 | 0.1 | 0.0 | 96.4 | 11.7 | 0.0 | 0.0 | 15.0 | 13.6 | 0.0 | 0.0 |
| | | AlexNet | xxiv | 95.5 | 3.6 | 2.4 | 3.5 | 85.7 | 3.6 | 2.5 | 3.5 | 34.6 | 2.5 | 2.5 | 2.5 | 77.9 | 0.0 | 0.0 | 2.6 | 72.2 | 0.0 | 0.0 | 2.6 | 12.9 | 0.0 | 0.0 | 2.5 |

Table 11: Image classification accuracy is very high with just the top 40 principal components both with 40 classes (≥ 85.7%; column v, for rows v–viii, xiii–xvi,and xxi–xxiv) and 1000 classes (≥72.2%; column xvii, for rows v–viii, xiii–xvi, and xxi–xxiv). This holds both for the 'Spampinato et al. (2017)' data, the 'Li et al. (2021) block' data, and the 'Li et al. (2021) randomized' data. This, together with Tables 9 and 10, demonstrates that they are giving class information implicitly to their training by use of image encoders pretrained on ImageNet, supports claim A, and calls claim I into question.

| | | 40 classes | | | | | | | | | | | | 1000 classes | | | | | | | | | | | |
| | | original | | | | top 40 | | | | bottom 960 | | | | original | | | | top 40 | | | | bottom 960 | | | |
| | | before | after joint pretraining | no pretraining | after separate | before | after joint pretraining | no pretraining | after separate | before | after joint pretraining | no pretraining | after separate | before | after joint pretraining | no pretraining | after separate | before | after joint pretraining | no pretraining | after separate | before | after joint pretraining | no pretraining | after separate |
| | | i | ii | iii | iv | v | vi | vii | viii | ix | x | xi | xii | xiii | xiv | xv | xvi | xvii | xviii | xix | xx | xxi | xxii | xxiii | xxiv |
|---|---|---|---|---|---|---|---|---|---|---|---|---|---|---|---|---|---|---|---|---|---|---|---|---|---|
| Spampinato et al. (2017) EEG | Inception-v3 | 2.6 | 3.0 | 2.5 | 55.5 | 2.2 | 2.6 | 2.5 | 55.5 | 2.5 | 3.1 | 2.5 | 2.3 | 0.1 | 0.1 | 0.1 | 55.5 | 0.1 | 0.1 | 0.1 | 55.5 | 0.1 | 0.1 | 0.1 | 2.3 |
| | ResNet-101 | 2.3 | 3.0 | 2.6 | 54.7 | 2.8 | 2.4 | 2.4 | 54.7 | 2.4 | 3.1 | 2.5 | 2.3 | 0.0 | 0.1 | 0.1 | 54.7 | 0.0 | 0.0 | 0.1 | 54.7 | 0.1 | 0.1 | 0.1 | 2.3 |
| | DenseNet-161 | 2.5 | 3.1 | 2.4 | 57.4 | 2.4 | 2.6 | 2.3 | 57.4 | 2.5 | 3.2 | 2.4 | 2.5 | 0.1 | 0.1 | 0.1 | 57.4 | 0.0 | 0.1 | 0.0 | 57.4 | 0.1 | 0.1 | 0.1 | 2.5 |
| | AlexNet | 2.3 | 2.5 | 2.4 | 57.3 | 2.4 | 2.6 | 2.6 | 57.3 | 2.4 | 2.5 | 2.6 | 2.6 | 0.1 | 0.1 | 0.1 | 57.3 | 0.0 | 0.1 | 0.3 | 57.3 | 0.1 | 0.1 | 0.1 | 2.5 |
| Spampinato et al. (2017) image | Inception-v3 | 99.7 | 43.2 | 2.7 | 3.4 | 99.4 | 13.5 | 2.9 | 2.2 | 45.5 | 29.4 | 2.7 | 2.5 | 92.8 | 16.5 | 0.0 | 0.0 | 98.9 | 4.0 | 0.0 | 0.0 | 12.8 | 9.3 | 0.0 | 0.0 |
| | ResNet-101 | 99.4 | 24.2 | 2.4 | 82.5 | 98.3 | 7.3 | 2.9 | 70.5 | 33.0 | 9.5 | 2.1 | 39.4 | 90.5 | 2.6 | 0.0 | 82.5 | 95.6 | 0.3 | 0.0 | 70.5 | 9.5 | 0.7 | 0.0 | 39.2 |
| | DenseNet-161 | 99.2 | 7.2 | 2.4 | 23.3 | 97.9 | 4.4 | 2.3 | 3.5 | 42.7 | 3.8 | 2.6 | 2.6 | 89.4 | 0.1 | 0.1 | 12.1 | 95.8 | 0.0 | 0.0 | 0.1 | 14.2 | 0.0 | 0.1 | 0.0 |
| | AlexNet | 95.8 | 2.6 | 2.7 | 18.7 | 85.8 | 2.6 | 2.7 | 19.1 | 34.1 | 2.6 | 2.4 | 2.4 | 78.4 | 0.0 | 0.3 | 18.7 | 72.5 | 0.0 | 0.0 | 19.1 | 12.5 | 0.0 | 0.0 | 2.4 |
| Li et al. (2021) block EEG | Inception-v3 | 2.4 | 6.8 | 1.9 | 77.0 | 1.8 | 7.1 | 2.0 | 69.8 | 2.6 | 3.6 | 2.5 | 32.1 | 0.0 | 0.4 | 0.0 | 77.0 | 0.0 | 0.2 | 0.1 | 69.8 | 0.0 | 0.2 | 0.1 | 28.0 |
| | ResNet-101 | 1.6 | 3.5 | 2.2 | 76.6 | 1.1 | 4.2 | 2.5 | 70.5 | 2.4 | 2.8 | 2.2 | 28.1 | 0.0 | 0.1 | 0.1 | 76.6 | 0.0 | 0.1 | 0.1 | 70.5 | 0.0 | 0.0 | 0.1 | 24.0 |
| | DenseNet-161 | 1.4 | 4.1 | 2.0 | 77.0 | 1.2 | 3.8 | 1.9 | 72.2 | 2.5 | 3.5 | 2.7 | 25.9 | 0.1 | 0.1 | 0.0 | 77.0 | 0.0 | 0.0 | 0.0 | 72.2 | 0.1 | 0.1 | 0.1 | 22.3 |
| | AlexNet | 2.9 | 4.1 | 2.9 | 76.5 | 2.5 | 4.2 | 2.2 | 70.0 | 2.0 | 3.4 | 2.7 | 27.8 | 0.0 | 0.1 | 0.1 | 76.4 | 0.0 | 0.3 | 0.2 | 70.0 | 0.0 | 0.0 | 0.1 | 24.2 |
| Li et al. (2021) block image | Inception-v3 | 99.8 | 96.4 | 3.0 | 9.3 | 99.4 | 94.7 | 3.1 | 2.9 | 46.0 | 46.8 | 2.4 | 2.5 | 93.0 | 75.1 | 0.1 | 0.5 | 98.8 | 87.0 | 0.0 | 0.0 | 13.2 | 11.7 | 0.0 | 0.0 |
| | ResNet-101 | 99.3 | 75.0 | 2.4 | 11.0 | 98.5 | 68.5 | 2.7 | 3.5 | 34.1 | 14.7 | 2.5 | 3.7 | 90.5 | 56.4 | 0.0 | 2.0 | 96.2 | 52.4 | 0.0 | 0.1 | 9.8 | 0.2 | 0.0 | 0.0 |
| | DenseNet-161 | 99.1 | 95.8 | 2.5 | 5.7 | 97.7 | 86.8 | 2.9 | 3.8 | 43.9 | 50.9 | 2.1 | 2.6 | 89.5 | 71.1 | 0.0 | 0.2 | 96.4 | 67.2 | 0.4 | 0.2 | 15.0 | 17.9 | 0.0 | 0.0 |
| | AlexNet | 95.5 | 60.9 | 2.5 | 3.5 | 85.7 | 50.9 | 2.4 | 3.2 | 34.6 | 15.6 | 2.5 | 2.3 | 77.9 | 25.4 | 0.0 | 2.6 | 72.2 | 19.7 | 0.0 | 2.5 | 12.9 | 4.2 | 0.0 | 2.3 |
| Li et al. (2021) randomized EEG | Inception-v3 | 2.6 | 2.5 | 2.5 | 1.6 | 2.6 | 2.9 | 2.6 | 2.5 | 3.0 | 2.2 | 2.7 | 1.6 | 0.0 | 0.1 | 0.1 | 1.5 | 0.1 | 0.1 | 0.1 | 2.5 | 0.2 | 0.1 | 0.2 | 1.4 |
| | ResNet-101 | 2.1 | 2.4 | 2.8 | 2.1 | 2.8 | 2.7 | 2.9 | 2.3 | 2.6 | 2.7 | 3.0 | 1.8 | 0.0 | 0.1 | 0.1 | 1.9 | 0.0 | 0.1 | 0.2 | 2.3 | 0.1 | 0.1 | 0.1 | 1.6 |
| | DenseNet-161 | 2.1 | 2.2 | 2.3 | 1.5 | 2.5 | 2.3 | 2.5 | 2.0 | 3.1 | 2.0 | 1.9 | 1.9 | 0.1 | 0.1 | 0.1 | 1.5 | 0.0 | 0.0 | 0.0 | 2.0 | 0.1 | 0.1 | 0.1 | 1.7 |
| | AlexNet | 2.5 | 2.8 | 3.0 | 2.2 | 2.2 | 2.4 | 2.9 | 1.6 | 2.6 | 2.8 | 3.1 | 2.4 | 0.2 | 0.1 | 0.2 | 2.2 | 0.3 | 0.0 | 0.1 | 1.6 | 0.1 | 0.1 | 0.1 | 1.9 |
| Li et al. (2021) randomized image | Inception-v3 | 99.8 | 91.1 | 3.4 | 9.4 | 99.4 | 84.7 | 3.0 | 2.9 | 46.0 | 57.1 | 2.1 | 2.5 | 93.0 | 64.0 | 0.1 | 0.0 | 98.8 | 73.3 | 0.0 | 0.0 | 13.2 | 20.4 | 0.0 | 0.0 |
| | ResNet-101 | 99.3 | 69.8 | 2.9 | 11.2 | 98.5 | 53.1 | 2.6 | 4.0 | 34.1 | 9.4 | 3.1 | 2.2 | 90.5 | 42.0 | 0.0 | 2.0 | 96.2 | 28.5 | 0.0 | 0.1 | 9.8 | 3.5 | 0.0 | 0.0 |
| | DenseNet-161 | 99.1 | 65.6 | 2.1 | 5.1 | 97.7 | 36.0 | 2.5 | 3.0 | 43.9 | 36.1 | 2.3 | 3.0 | 89.5 | 30.5 | 0.1 | 2.0 | 96.4 | 11.7 | 0.0 | 0.0 | 15.0 | 13.6 | 0.0 | 0.0 |
| | AlexNet | 95.5 | 3.6 | 2.4 | 3.5 | 85.7 | 3.6 | 2.5 | 3.5 | 34.6 | 2.5 | 2.5 | 2.5 | 77.9 | 0.0 | 0.0 | 2.6 | 72.2 | 0.0 | 0.0 | 2.6 | 12.9 | 0.0 | 0.0 | 2.5 |

Table 12: Image classification accuracy is much lower than perfect (≤46.0%) before training when discarding the top 40 principal components (columns ix and xxi, for rows v–viii, xiii–xvi, and xxi–xxiv). This holds both for 40 classes and 1000 classes and both for the 'Spampinato et al. (2017)' data, the 'Li et al. (2021) block' data, and the 'Li et al. (2021) randomized' data. Thus there is much less class information outside the top 40 principal components. This demonstrates that they are giving little more than class information implicitly to their training by use of image encoders pretrained on ImageNet and further supports claim A and calls claim I into question.

Column grouping: **40 classes** — original (i–iv), top 40 (v–viii), bottom 960 (ix–xii); **1000 classes** — original (xiii–xvi), top 40 (xvii–xx), bottom 960 (xxi–xxiv). Within each block the sub-columns are: before; after joint (pretraining / no pretraining); after separate.

| Dataset | Mod. | Model | # | i | ii | iii | iv | v | vi | vii | viii | ix | x | xi | xii | xiii | xiv | xv | xvi | xvii | xviii | xix | xx | xxi | xxii | xxiii | xxiv |
|---|---|---|---|---|---|---|---|---|---|---|---|---|---|---|---|---|---|---|---|---|---|---|---|---|---|---|---|
| Spampinato et al. (2017) | EEG | Inception-v3 | i | 2.6 | 3.0 | 2.5 | 55.5 | 2.2 | 2.6 | 2.5 | 55.5 | 2.5 | 3.1 | 2.5 | 2.3 | 0.1 | 0.1 | 0.1 | 55.5 | 0.1 | 0.1 | 0.1 | 55.5 | 0.1 | 0.1 | 0.1 | 2.3 |
| | | ResNet-101 | ii | 2.3 | 3.0 | 2.6 | 54.7 | 2.8 | 2.4 | 2.4 | 54.7 | 2.4 | 3.1 | 2.5 | 2.3 | 0.0 | 0.1 | 0.1 | 54.7 | 0.0 | 0.0 | 0.1 | 54.7 | 0.1 | 0.1 | 0.1 | 2.3 |
| | | DenseNet-161 | iii | 2.5 | 3.1 | 2.4 | 57.4 | 2.4 | 2.6 | 2.3 | 57.4 | 2.5 | 3.2 | 2.4 | 2.5 | 0.1 | 0.1 | 0.0 | 57.4 | 0.0 | 0.1 | 0.0 | 57.4 | 0.1 | 0.1 | 0.1 | 2.5 |
| | | AlexNet | iv | 2.3 | 2.5 | 2.4 | 57.3 | 2.4 | 2.6 | 2.6 | 57.3 | 2.4 | 2.5 | 2.5 | 2.5 | 0.1 | 0.1 | 0.1 | 57.3 | 0.0 | 0.1 | 0.1 | 57.3 | 0.1 | 0.1 | 0.1 | 2.5 |
| | image | Inception-v3 | v | 99.7 | 43.2 | 2.7 | 3.4 | 99.4 | 13.5 | 2.9 | 2.2 | 45.5 | 29.4 | 2.7 | 2.5 | 92.8 | 16.5 | 0.1 | 0.0 | 98.9 | 4.0 | 0.0 | 0.0 | 12.8 | 9.3 | 0.0 | 0.0 |
| | | ResNet-101 | vi | 99.4 | 24.2 | 2.4 | 82.5 | 98.3 | 7.3 | 2.9 | 70.5 | 33.0 | 9.5 | 2.1 | 39.4 | 90.5 | 2.6 | 0.0 | 82.5 | 95.6 | 0.3 | 0.0 | 70.5 | 9.5 | 0.7 | 0.0 | 39.2 |
| | | DenseNet-161 | vii | 99.2 | 7.2 | 2.4 | 23.3 | 97.9 | 4.4 | 2.3 | 3.5 | 42.7 | 3.8 | 2.6 | 2.6 | 89.4 | 0.1 | 0.0 | 12.1 | 95.8 | 0.0 | 0.0 | 0.1 | 14.2 | 0.0 | 0.1 | 0.0 |
| | | AlexNet | viii | 95.8 | 2.6 | 2.7 | 18.7 | 85.8 | 2.6 | 2.7 | 19.1 | 34.1 | 2.6 | 2.4 | 2.4 | 78.4 | 0.0 | 0.3 | 18.7 | 72.5 | 0.2 | 0.3 | 19.1 | 12.5 | 0.0 | 0.0 | 2.4 |
| Li et al. (2021) Block | EEG | Inception-v3 | ix | 2.4 | 6.8 | 1.9 | 77.0 | 1.8 | 7.1 | 2.0 | 69.8 | 2.6 | 3.6 | 2.5 | 32.1 | 0.1 | 0.4 | 0.0 | 77.0 | 0.0 | 0.2 | 0.1 | 69.8 | 0.0 | 0.2 | 0.1 | 28.0 |
| | | ResNet-101 | x | 1.6 | 3.5 | 2.2 | 76.6 | 1.1 | 4.2 | 2.5 | 70.5 | 2.4 | 2.8 | 2.2 | 28.1 | 0.0 | 0.1 | 0.1 | 76.6 | 0.0 | 0.1 | 0.1 | 70.5 | 0.0 | 0.0 | 0.1 | 24.0 |
| | | DenseNet-161 | xi | 1.4 | 4.1 | 2.0 | 77.0 | 1.2 | 3.8 | 1.9 | 77.0 | 2.5 | 3.5 | 2.7 | 25.9 | 0.1 | 0.1 | 0.0 | 77.0 | 0.0 | 0.0 | 0.0 | 72.2 | 0.1 | 0.1 | 0.1 | 22.3 |
| | | AlexNet | xii | 2.9 | 4.1 | 2.9 | 76.5 | 2.5 | 4.2 | 2.2 | 70.0 | 2.0 | 3.4 | 2.5 | 27.8 | 0.0 | 0.1 | 0.1 | 76.4 | 0.0 | 0.3 | 0.2 | 70.0 | 0.0 | 0.0 | 0.1 | 24.2 |
| | image | Inception-v3 | xiii | 99.8 | 96.4 | 3.0 | 9.3 | 99.4 | 94.7 | 3.1 | 2.9 | 46.0 | 46.8 | 2.4 | 2.5 | 93.0 | 75.1 | 0.1 | 0.5 | 98.8 | 87.0 | 0.0 | 0.0 | 13.2 | 11.7 | 0.0 | 0.0 |
| | | ResNet-101 | xiv | 99.3 | 75.0 | 2.4 | 11.0 | 98.5 | 68.5 | 2.7 | 3.5 | 34.1 | 14.7 | 2.5 | 3.7 | 90.5 | 56.4 | 0.0 | 3.7 | 96.2 | 52.4 | 0.1 | 0.1 | 9.8 | 0.2 | 0.0 | 0.0 |
| | | DenseNet-161 | xv | 99.1 | 95.8 | 2.5 | 5.7 | 97.7 | 86.8 | 2.9 | 3.8 | 43.9 | 50.9 | 2.1 | 2.6 | 89.5 | 71.1 | 0.4 | 0.2 | 96.4 | 67.2 | 0.4 | 0.2 | 15.0 | 17.9 | 0.0 | 0.0 |
| | | AlexNet | xvi | 95.5 | 60.9 | 2.5 | 3.5 | 85.7 | 50.9 | 2.4 | 3.2 | 34.6 | 15.6 | 2.5 | 2.3 | 77.9 | 25.4 | 0.0 | 2.6 | 72.2 | 19.7 | 0.0 | 2.5 | 12.9 | 4.2 | 0.0 | 2.3 |
| Li et al. (2021) randomized | EEG | Inception-v3 | xvii | 2.6 | 2.5 | 2.5 | 1.6 | 2.6 | 2.9 | 2.6 | 2.5 | 3.0 | 2.2 | 2.7 | 1.6 | 0.1 | 0.1 | 0.1 | 1.5 | 0.1 | 0.1 | 0.0 | 2.5 | 0.2 | 0.1 | 0.2 | 1.4 |
| | | ResNet-101 | xviii | 2.6 | 2.4 | 2.8 | 2.1 | 2.8 | 2.7 | 2.9 | 2.3 | 2.6 | 2.7 | 3.0 | 1.8 | 0.0 | 0.1 | 0.2 | 1.9 | 0.0 | 0.1 | 0.2 | 2.3 | 0.1 | 0.2 | 0.1 | 1.6 |
| | | DenseNet-161 | xix | 2.1 | 2.2 | 2.3 | 1.5 | 2.5 | 2.3 | 2.5 | 2.0 | 3.1 | 2.0 | 1.9 | 1.9 | 0.1 | 0.1 | 0.1 | 1.5 | 0.0 | 0.0 | 0.1 | 2.0 | 0.1 | 0.1 | 0.1 | 1.7 |
| | | AlexNet | xx | 2.5 | 2.8 | 3.0 | 2.2 | 2.2 | 2.4 | 2.9 | 1.6 | 2.6 | 2.8 | 3.1 | 2.4 | 0.2 | 0.1 | 0.2 | 2.2 | 0.3 | 0.0 | 0.1 | 1.6 | 0.1 | 0.1 | 0.1 | 1.9 |
| | image | Inception-v3 | xxi | 99.8 | 91.1 | 3.4 | 9.4 | 99.4 | 84.7 | 3.0 | 2.9 | 46.0 | 57.1 | 2.1 | 2.5 | 93.0 | 64.0 | 0.1 | 0.6 | 98.8 | 73.3 | 0.0 | 0.0 | 13.2 | 20.4 | 0.0 | 0.0 |
| | | ResNet-101 | xxii | 99.3 | 69.8 | 2.9 | 11.2 | 98.5 | 53.1 | 2.6 | 4.0 | 34.1 | 9.4 | 2.3 | 2.2 | 90.5 | 42.0 | 0.0 | 2.0 | 96.2 | 28.5 | 0.1 | 0.1 | 9.8 | 3.5 | 0.0 | 0.0 |
| | | DenseNet-161 | xxiii | 99.1 | 65.6 | 2.1 | 5.1 | 97.7 | 36.0 | 2.5 | 3.0 | 43.9 | 36.1 | 2.3 | 3.0 | 89.5 | 30.5 | 0.1 | 0.0 | 96.4 | 11.7 | 0.0 | 0.0 | 15.0 | 13.6 | 0.0 | 0.0 |
| | | AlexNet | xxiv | 95.5 | 3.6 | 2.4 | 3.5 | 85.7 | 3.6 | 2.5 | 3.5 | 34.6 | 2.5 | 2.5 | 2.5 | 77.9 | 0.0 | 0.0 | 2.6 | 72.2 | 0.0 | 0.0 | 2.6 | 12.9 | 0.0 | 0.2 | 2.5 |

Table 13: With only five exceptions, all in the bottom 960 principal components, image classification accuracy decreases after joint training with pretraining (compare columns ii, vi, x, xiv, xviii, and xxii to i, v, ix, xiii, xvii, and xxi, respectively, for rows v–viii, xiii–xvi, and xxi–xxiv). This holds both for 40 classes and 1000 classes, both for the 'Spampinato et al. (2017)' data, and the 'Li et al. (2021) block' data, and the 'Li et al. (2021) randomized' data, and whether using all components or just the 40 principal components. This suggests that joint training is hurting, not helping, and supports claim C(2).

The table is organized into six column-groups of four sub-columns each. Groups 1–3 correspond to **40 classes** (original, top 40, bottom 960); groups 4–6 correspond to **1000 classes** (original, top 40, bottom 960). Within each group the four sub-columns are: **before**, **after joint (pretraining)**, **after joint (no pretraining)**, **after separate**.

| Dataset | Model | # | i | ii | iii | iv | v | vi | vii | viii | ix | x | xi | xii | xiii | xiv | xv | xvi | xvii | xviii | xix | xx | xxi | xxii | xxiii | xxiv |
|---|---|---|---|---|---|---|---|---|---|---|---|---|---|---|---|---|---|---|---|---|---|---|---|---|---|---|
| Spampinato et al. (2017) EEG | Inception-v3 | i | 2.6 | 3.0 | 2.5 | 55.5 | 2.2 | 2.6 | 2.5 | 55.5 | 2.5 | 3.1 | 2.5 | 2.5 | 0.1 | 0.1 | 0.1 | 55.5 | 0.1 | 0.1 | 0.1 | 0.1 | 0.1 | 0.1 | 0.1 | 2.3 |
| | ResNet-101 | ii | 2.3 | 3.0 | 2.6 | 54.7 | 2.8 | 2.4 | 2.4 | 54.7 | 2.4 | 3.1 | 2.5 | 2.3 | 0.0 | 0.1 | 0.1 | 54.7 | 0.0 | 0.0 | 0.1 | 0.1 | 0.1 | 0.1 | 0.1 | 2.3 |
| | DenseNet-161 | iii | 2.5 | 3.1 | 2.4 | 57.4 | 2.4 | 2.6 | 2.3 | 57.4 | 2.5 | 3.2 | 2.4 | 2.5 | 0.1 | 0.1 | 0.1 | 57.4 | 0.0 | 0.1 | 0.0 | 0.0 | 0.1 | 0.1 | 0.1 | 2.5 |
| | AlexNet | iv | 2.3 | 2.5 | 2.4 | 57.3 | 2.4 | 2.6 | 2.6 | 57.3 | 2.4 | 2.5 | 2.6 | 2.6 | 0.1 | 0.1 | 0.1 | 57.3 | 0.0 | 0.1 | 0.1 | 0.1 | 0.1 | 0.1 | 0.1 | 2.5 |
| Spampinato et al. (2017) image | Inception-v3 | v | 99.7 | 43.2 | 2.7 | 3.4 | 99.4 | 13.5 | 2.9 | 2.2 | 45.5 | 29.4 | 2.7 | 2.5 | 92.8 | 16.5 | 0.1 | 0.0 | 98.9 | 4.0 | 0.0 | 0.0 | 12.8 | 9.3 | 0.0 | 0.0 |
| | ResNet-101 | vi | 99.4 | 24.2 | 2.4 | 82.5 | 98.3 | 7.3 | 2.9 | 70.5 | 33.0 | 9.5 | 2.1 | 39.4 | 90.5 | 2.6 | 0.0 | 82.5 | 95.6 | 0.3 | 0.0 | 70.5 | 9.5 | 0.7 | 0.0 | 39.2 |
| | DenseNet-161 | vii | 99.2 | 7.2 | 2.4 | 23.3 | 97.9 | 4.4 | 2.3 | 3.5 | 42.7 | 3.8 | 2.6 | 2.6 | 89.4 | 0.1 | 0.0 | 12.1 | 95.8 | 0.0 | 0.0 | 0.1 | 14.2 | 0.0 | 0.1 | 0.0 |
| | AlexNet | viii | 95.8 | 2.6 | 2.7 | 18.7 | 85.8 | 2.6 | 2.7 | 19.1 | 34.1 | 2.6 | 2.4 | 2.4 | 78.4 | 0.0 | 0.3 | 18.7 | 72.5 | 0.0 | 0.3 | 19.1 | 12.5 | 0.0 | 0.0 | 2.4 |
| Li et al. (2021) block EEG | Inception-v3 | ix | 2.4 | 6.8 | 1.9 | 77.0 | 1.8 | 7.1 | 2.0 | 69.8 | 2.6 | 3.6 | 2.5 | 32.1 | 0.1 | 0.4 | 0.1 | 77.0 | 0.0 | 0.2 | 0.1 | 69.8 | 0.0 | 0.2 | 0.1 | 28.0 |
| | ResNet-101 | x | 1.6 | 3.5 | 2.2 | 76.6 | 1.1 | 4.2 | 2.5 | 70.5 | 2.4 | 2.8 | 2.2 | 28.1 | 0.0 | 0.1 | 0.1 | 76.6 | 0.0 | 0.0 | 0.1 | 70.5 | 0.0 | 0.0 | 0.1 | 24.0 |
| | DenseNet-161 | xi | 1.4 | 4.1 | 2.0 | 77.0 | 1.2 | 3.8 | 1.9 | 72.2 | 2.5 | 3.5 | 2.7 | 25.9 | 0.1 | 0.1 | 0.1 | 77.0 | 0.0 | 0.0 | 0.2 | 72.2 | 0.1 | 0.1 | 0.1 | 22.3 |
| | AlexNet | xii | 2.9 | 4.1 | 2.9 | 76.5 | 2.5 | 4.2 | 2.2 | 70.0 | 2.0 | 3.4 | 2.7 | 27.8 | 0.0 | 0.1 | 0.1 | 76.4 | 0.0 | 0.3 | 0.2 | 70.0 | 0.0 | 0.0 | 0.1 | 24.2 |
| Li et al. (2021) block image | Inception-v3 | xiii | 99.8 | 96.4 | 3.0 | 9.3 | 99.4 | 94.7 | 3.1 | 2.9 | 46.0 | 46.8 | 2.4 | 93.0 | 93.0 | 75.1 | 0.1 | 0.5 | 98.8 | 87.0 | 0.0 | 0.0 | 13.2 | 11.7 | 0.0 | 0.0 |
| | ResNet-101 | xiv | 99.3 | 75.0 | 2.4 | 11.0 | 98.5 | 68.5 | 2.7 | 3.5 | 34.1 | 14.7 | 2.5 | 90.5 | 90.5 | 56.4 | 0.1 | 2.0 | 96.2 | 52.4 | 0.1 | 0.1 | 9.8 | 0.2 | 0.0 | 0.0 |
| | DenseNet-161 | xv | 99.1 | 95.8 | 2.5 | 5.7 | 97.7 | 86.8 | 2.9 | 3.8 | 43.9 | 50.9 | 2.1 | 89.5 | 89.5 | 71.1 | 0.1 | 0.2 | 96.4 | 67.2 | 0.4 | 0.2 | 15.0 | 17.9 | 0.0 | 0.0 |
| | AlexNet | xvi | 95.5 | 60.9 | 2.5 | 3.5 | 85.7 | 50.9 | 2.4 | 3.2 | 34.6 | 15.6 | 2.5 | 77.9 | 77.9 | 25.4 | 0.0 | 2.6 | 72.2 | 19.7 | 0.0 | 2.5 | 12.9 | 4.2 | 0.0 | 2.3 |
| Li et al. (2021) randomized EEG | Inception-v3 | xvii | 2.6 | 2.5 | 2.5 | 1.6 | 2.6 | 2.9 | 2.6 | 2.5 | 3.0 | 2.2 | 2.7 | 1.6 | 0.1 | 0.1 | 0.1 | 1.5 | 0.1 | 0.1 | 0.0 | 0.2 | 0.2 | 0.1 | 0.2 | 1.4 |
| | ResNet-101 | xviii | 2.6 | 2.4 | 2.8 | 2.1 | 2.8 | 2.7 | 2.9 | 2.3 | 2.6 | 2.7 | 3.0 | 1.8 | 0.0 | 0.1 | 0.1 | 1.9 | 0.0 | 0.1 | 0.2 | 2.3 | 0.1 | 0.2 | 0.1 | 1.6 |
| | DenseNet-161 | xix | 2.1 | 2.2 | 2.3 | 1.5 | 2.5 | 2.3 | 2.5 | 2.0 | 3.1 | 2.0 | 1.9 | 1.9 | 0.1 | 0.1 | 0.1 | 1.5 | 0.0 | 0.0 | 0.1 | 2.0 | 0.1 | 0.1 | 0.1 | 1.7 |
| | AlexNet | xx | 2.5 | 2.8 | 3.0 | 2.2 | 2.2 | 2.4 | 2.9 | 1.6 | 2.6 | 2.8 | 3.1 | 2.4 | 0.2 | 0.1 | 0.2 | 2.2 | 0.3 | 0.0 | 0.1 | 1.6 | 0.1 | 0.1 | 0.1 | 1.9 |
| Li et al. (2021) randomized image | Inception-v3 | xxi | 99.8 | 91.1 | 3.4 | 9.4 | 99.4 | 84.7 | 3.0 | 2.9 | 46.0 | 57.1 | 2.1 | 2.5 | 93.0 | 64.0 | 0.1 | 0.0 | 98.8 | 73.3 | 0.0 | 0.0 | 13.2 | 20.4 | 0.0 | 0.0 |
| | ResNet-101 | xxii | 99.3 | 69.8 | 2.9 | 11.2 | 98.5 | 53.1 | 2.6 | 4.0 | 34.1 | 9.4 | 3.1 | 2.2 | 90.5 | 42.0 | 0.0 | 2.0 | 96.2 | 28.5 | 0.0 | 0.1 | 9.8 | 3.5 | 0.0 | 0.0 |
| | DenseNet-161 | xxiii | 99.1 | 65.6 | 2.1 | 5.1 | 97.7 | 36.0 | 2.5 | 3.0 | 43.9 | 36.1 | 2.3 | 3.0 | 89.5 | 30.5 | 0.1 | 3.0 | 96.4 | 11.7 | 0.0 | 0.0 | 15.0 | 13.6 | 0.0 | 0.0 |
| | AlexNet | xxiv | 95.5 | 3.6 | 2.4 | 3.5 | 85.7 | 3.6 | 2.5 | 3.5 | 34.6 | 2.5 | 2.5 | 2.5 | 77.9 | 0.0 | 0.0 | 2.6 | 72.2 | 0.0 | 0.0 | 2.6 | 12.9 | 0.0 | 0.0 | 2.5 |

Table 14: With the exception of a small number of cases that are marginally above chance, EEG classification accuracy is at chance except after separate training on confounded data either on all components or the top 40 components (columns iv, viii, xvi, and xx, for rows i–iv and ix–xii). This, together with Table 15, suggests that separate training is able to extract class from confounded data but not nonconfounded data and that joint training is not able to extract class from any data, and supports claims C(1) and D.

**40 classes** — original: before (i), after joint pretraining (ii), after joint no pretraining (iii), after separate (iv); top 40: before (v), after joint pretraining (vi), after joint no pretraining (vii), after separate (viii); bottom 960: before (ix), after joint pretraining (x), after joint no pretraining (xi), after separate (xii).
**1000 classes** — original: before (xiii), after joint pretraining (xiv), after joint no pretraining (xv), after separate (xvi); top 40: before (xvii), after joint pretraining (xviii), after joint no pretraining (xix), after separate (xx); bottom 960: before (xxi), after joint pretraining (xxii), after joint no pretraining (xxiii), after separate (xxiv).

| Study | Type | Model | # | i | ii | iii | iv | v | vi | vii | viii | ix | x | xi | xii | xiii | xiv | xv | xvi | xvii | xviii | xix | xx | xxi | xxii | xxiii | xxiv |
|---|---|---|---|---|---|---|---|---|---|---|---|---|---|---|---|---|---|---|---|---|---|---|---|---|---|---|---|
| Spampinato et al. (2017) | EEG | Inception-v3 | i | 2.6 | 3.0 | 2.5 | 55.5 | 2.2 | 2.6 | 2.5 | 55.5 | 2.5 | 3.1 | 2.5 | 2.3 | 0.1 | 0.1 | 0.1 | 55.5 | 0.1 | 0.1 | 0.1 | 55.5 | 0.1 | 0.1 | 0.1 | 2.3 |
| | | ResNet-101 | ii | 2.3 | 3.0 | 2.6 | 54.7 | 2.8 | 2.6 | 2.4 | 54.7 | 2.4 | 3.1 | 2.5 | 2.3 | 0.0 | 0.1 | 0.1 | 54.7 | 0.0 | 0.0 | 0.1 | 54.7 | 0.1 | 0.1 | 0.1 | 2.3 |
| | | DenseNet-161 | iii | 2.5 | 3.1 | 2.4 | 57.4 | 2.4 | 2.6 | 2.3 | 57.4 | 2.5 | 3.2 | 2.4 | 2.5 | 0.1 | 0.1 | 0.1 | 57.4 | 0.0 | 0.1 | 0.0 | 57.4 | 0.1 | 0.1 | 0.1 | 2.5 |
| | | AlexNet | iv | 2.3 | 2.5 | 2.4 | 57.3 | 2.4 | 2.6 | 2.6 | 57.3 | 2.4 | 2.5 | 2.6 | 2.5 | 0.1 | 0.1 | 0.1 | 57.3 | 0.0 | 0.1 | 0.1 | 57.3 | 0.1 | 0.1 | 0.1 | 2.5 |
| | image | Inception-v3 | v | 99.7 | 43.2 | 2.7 | 3.4 | 99.4 | 13.5 | 2.9 | 2.2 | 45.5 | 29.4 | 2.7 | 2.5 | 92.8 | 16.5 | 0.1 | 0.0 | 98.9 | 4.0 | 0.0 | 0.0 | 12.8 | 9.3 | 0.0 | 0.0 |
| | | ResNet-101 | vi | 99.4 | 24.2 | 2.4 | 82.5 | 98.3 | 7.3 | 2.9 | 70.5 | 33.0 | 9.5 | 2.1 | 39.4 | 90.5 | 2.6 | 0.0 | 82.5 | 95.6 | 0.3 | 0.0 | 70.5 | 9.5 | 0.7 | 0.0 | 39.2 |
| | | DenseNet-161 | vii | 99.2 | 7.2 | 2.4 | 23.3 | 97.9 | 4.4 | 2.3 | 3.5 | 42.7 | 3.8 | 2.6 | 2.6 | 89.4 | 0.1 | 0.0 | 12.1 | 95.8 | 0.0 | 0.0 | 0.1 | 14.2 | 0.0 | 0.1 | 0.0 |
| | | AlexNet | viii | 95.8 | 2.6 | 2.7 | 18.7 | 85.8 | 2.6 | 2.7 | 19.1 | 34.1 | 2.6 | 2.4 | 2.4 | 78.4 | 0.0 | 0.3 | 18.7 | 72.5 | 0.0 | 0.3 | 19.1 | 12.5 | 0.0 | 0.0 | 2.4 |
| Li et al. (2021) Block | EEG | Inception-v3 | ix | 2.4 | 6.8 | 1.9 | 77.0 | 1.8 | 7.1 | 2.0 | 69.8 | 2.6 | 3.6 | 2.5 | 32.1 | 0.1 | 0.4 | 0.0 | 77.0 | 0.0 | 0.2 | 0.1 | 69.8 | 0.0 | 0.2 | 0.1 | 28.0 |
| | | ResNet-101 | x | 1.6 | 3.5 | 2.2 | 76.6 | 1.1 | 4.2 | 2.5 | 70.5 | 2.4 | 2.8 | 2.2 | 28.1 | 0.0 | 0.1 | 0.1 | 76.6 | 0.0 | 0.0 | 0.1 | 70.5 | 0.0 | 0.0 | 0.1 | 24.0 |
| | | DenseNet-161 | xi | 1.4 | 4.1 | 2.0 | 77.0 | 1.2 | 3.8 | 1.9 | 72.2 | 2.5 | 3.5 | 2.7 | 25.9 | 0.1 | 0.1 | 0.0 | 77.0 | 0.0 | 0.0 | 0.0 | 72.2 | 0.0 | 0.1 | 0.1 | 22.3 |
| | | AlexNet | xii | 2.9 | 4.1 | 2.9 | 76.5 | 2.5 | 4.2 | 2.2 | 70.0 | 2.0 | 3.4 | 2.2 | 27.8 | 0.1 | 0.1 | 0.1 | 76.4 | 0.0 | 0.3 | 0.2 | 70.0 | 0.0 | 0.2 | 0.2 | 24.2 |
| | image | Inception-v3 | xiii | 99.8 | 96.4 | 3.0 | 9.3 | 99.4 | 94.7 | 3.1 | 2.9 | 46.0 | 46.8 | 2.4 | 2.9 | 93.0 | 75.1 | 0.1 | 0.5 | 98.8 | 87.0 | 0.0 | 0.1 | 13.2 | 11.7 | 0.0 | 0.0 |
| | | ResNet-101 | xiv | 99.3 | 75.0 | 2.4 | 11.0 | 98.5 | 68.5 | 2.7 | 3.5 | 34.1 | 14.7 | 2.5 | 3.7 | 90.5 | 56.4 | 0.0 | 2.0 | 96.2 | 52.4 | 0.0 | 0.2 | 9.8 | 0.2 | 0.0 | 0.0 |
| | | DenseNet-161 | xv | 99.1 | 95.8 | 2.5 | 5.7 | 97.7 | 86.8 | 2.9 | 3.8 | 43.9 | 50.9 | 2.1 | 2.6 | 89.5 | 71.1 | 0.0 | 0.2 | 96.4 | 67.2 | 0.0 | 0.4 | 15.0 | 17.9 | 0.0 | 0.0 |
| | | AlexNet | xvi | 95.5 | 60.9 | 2.5 | 3.5 | 85.7 | 50.9 | 2.4 | 3.2 | 34.6 | 15.6 | 2.5 | 2.3 | 77.9 | 25.4 | 0.0 | 2.6 | 72.2 | 19.7 | 0.0 | 0.0 | 12.9 | 4.2 | 0.0 | 2.3 |
| Li et al. (2021) randomized | EEG | Inception-v3 | xvii | 2.6 | 2.5 | 2.5 | 1.6 | 2.6 | 2.9 | 2.6 | 2.5 | 3.0 | 2.2 | 2.7 | 1.6 | 0.1 | 0.1 | 0.1 | 0.1 | 0.1 | 0.1 | 0.0 | 2.5 | 0.2 | 0.1 | 0.0 | 1.4 |
| | | ResNet-101 | xviii | 2.1 | 2.4 | 2.8 | 2.1 | 2.8 | 2.7 | 2.9 | 2.3 | 2.6 | 2.7 | 3.0 | 1.8 | 0.0 | 0.1 | 0.1 | 1.9 | 0.0 | 0.1 | 0.2 | 2.3 | 0.1 | 0.2 | 0.1 | 1.6 |
| | | DenseNet-161 | xix | 2.1 | 2.2 | 2.3 | 1.5 | 2.5 | 2.3 | 2.5 | 2.0 | 3.1 | 2.0 | 1.9 | 1.9 | 0.1 | 0.1 | 0.1 | 1.5 | 0.0 | 0.0 | 0.0 | 2.0 | 0.1 | 0.1 | 0.0 | 1.7 |
| | | AlexNet | xx | 2.5 | 2.8 | 3.0 | 2.2 | 2.2 | 2.4 | 2.9 | 1.6 | 2.6 | 2.8 | 3.1 | 2.4 | 0.2 | 0.1 | 0.2 | 2.2 | 0.3 | 0.0 | 0.1 | 1.6 | 0.1 | 0.1 | 0.0 | 1.9 |
| | image | Inception-v3 | xxi | 99.8 | 91.1 | 3.4 | 9.4 | 99.4 | 84.7 | 3.0 | 2.9 | 46.0 | 57.1 | 2.1 | 2.5 | 93.0 | 64.0 | 0.1 | 0.6 | 98.8 | 73.3 | 0.0 | 0.0 | 13.2 | 20.4 | 0.0 | 0.0 |
| | | ResNet-101 | xxii | 99.3 | 69.8 | 2.9 | 11.2 | 98.5 | 53.1 | 2.6 | 4.0 | 34.1 | 9.4 | 3.1 | 2.2 | 90.5 | 42.0 | 0.0 | 2.0 | 96.2 | 28.5 | 0.0 | 0.1 | 9.8 | 3.5 | 0.0 | 0.0 |
| | | DenseNet-161 | xxiii | 99.1 | 65.6 | 2.1 | 5.1 | 97.7 | 36.0 | 2.5 | 3.0 | 43.9 | 36.1 | 2.3 | 3.0 | 89.5 | 30.5 | 0.1 | 0.0 | 96.4 | 11.7 | 0.0 | 0.0 | 15.0 | 13.6 | 0.0 | 0.0 |
| | | AlexNet | xxiv | 95.5 | 3.6 | 2.4 | 3.5 | 85.7 | 3.6 | 2.5 | 3.5 | 34.6 | 2.5 | 2.5 | 2.5 | 77.9 | 0.0 | 0.0 | 2.6 | 72.2 | 0.0 | 0.0 | 2.6 | 12.9 | 0.0 | 0.0 | 2.5 |

Table 15: EEG classification accuracy is also above chance after separate training on the 'Li et al. (2021) block' data in the bottom 960 components (columns xii and xxiv, for rows ix—xii). This, together with Table 14, suggests that separate training is able to extract class from confounded data but not nonconfounded data and that joint training is not able to extract class from any data, and supports claims C(1) and D.

Column groups (each group: *before* | *after joint — pretraining* | *after joint — no pretraining* | *after separate*):
- 40 classes — original: i–iv; top 40: v–viii; bottom 960: ix–xii
- 1000 classes — original: xiii–xvi; top 40: xvii–xx; bottom 960: xxi–xxiv

| Dataset | | Network | # | i | ii | iii | iv | v | vi | vii | viii | ix | x | xi | xii | xiii | xiv | xv | xvi | xvii | xviii | xix | xx | xxi | xxii | xxiii | xxiv |
|---|---|---|---|---|---|---|---|---|---|---|---|---|---|---|---|---|---|---|---|---|---|---|---|---|---|---|---|
| Spampinato et al. (2017) | EEG | Inception-v3 | i | 2.6 | 3.0 | 2.5 | 55.5 | 2.2 | 2.6 | 2.5 | 55.5 | 2.5 | 3.1 | 2.5 | 2.3 | 0.1 | 0.1 | 0.1 | 55.5 | 0.1 | 0.1 | 0.1 | 55.5 | 0.1 | 0.1 | 0.1 | 2.3 |
| | | ResNet-101 | ii | 2.3 | 3.0 | 2.4 | 54.7 | 2.8 | 2.6 | 2.4 | 54.7 | 2.4 | 3.1 | 2.5 | 2.3 | 0.0 | 0.1 | 0.1 | 54.7 | 0.0 | 0.0 | 0.1 | 54.7 | 0.1 | 0.1 | 0.1 | 2.3 |
| | | DenseNet-161 | iii | 2.5 | 3.1 | 2.4 | 57.4 | 2.4 | 2.6 | 2.3 | 57.4 | 2.5 | 3.2 | 2.4 | 2.5 | 0.1 | 0.1 | 0.1 | 57.4 | 0.0 | 0.1 | 0.0 | 57.4 | 0.1 | 0.1 | 0.1 | 2.5 |
| | | AlexNet | iv | 2.3 | 2.5 | 2.4 | 57.3 | 2.4 | 2.6 | 2.6 | 57.3 | 2.4 | 2.5 | 2.6 | 2.5 | 0.1 | 0.1 | 0.1 | 57.3 | 0.0 | 0.1 | 0.1 | 57.3 | 0.1 | 0.1 | 0.1 | 2.5 |
| | image | Inception-v3 | v | 99.7 | 43.2 | 2.7 | 3.4 | 99.4 | 13.5 | 2.9 | 2.2 | 45.5 | 29.4 | 2.7 | 2.5 | 92.8 | 16.5 | 0.1 | 0.0 | 98.9 | 4.0 | 0.0 | 0.0 | 12.8 | 9.3 | 0.0 | 0.0 |
| | | ResNet-101 | vi | 99.4 | 24.2 | 2.4 | 82.5 | 98.3 | 7.3 | 2.9 | 70.5 | 33.0 | 9.5 | 2.1 | 39.4 | 90.5 | 2.6 | 0.0 | 82.5 | 95.6 | 0.3 | 0.0 | 70.5 | 9.5 | 0.7 | 0.0 | 39.2 |
| | | DenseNet-161 | vii | 99.2 | 7.2 | 2.4 | 23.3 | 97.9 | 4.4 | 2.3 | 3.5 | 42.7 | 3.8 | 2.6 | 2.6 | 89.4 | 0.1 | 0.0 | 12.1 | 95.8 | 0.0 | 0.0 | 0.1 | 14.2 | 0.0 | 0.1 | 0.0 |
| | | AlexNet | viii | 95.8 | 2.6 | 2.7 | 18.7 | 85.8 | 2.6 | 2.7 | 19.1 | 34.1 | 2.6 | 2.4 | 2.4 | 78.4 | 0.0 | 0.3 | 18.7 | 72.5 | 0.0 | 0.3 | 19.1 | 12.5 | 0.0 | 0.0 | 2.4 |
| Li et al. (2021) Block | EEG | Inception-v3 | ix | 2.4 | 6.8 | 1.9 | 77.0 | 1.8 | 7.1 | 2.0 | 69.8 | 2.6 | 3.6 | 2.5 | 32.1 | 0.1 | 0.4 | 0.0 | 77.0 | 0.0 | 0.2 | 0.1 | 69.8 | 0.0 | 0.2 | 0.1 | 28.0 |
| | | ResNet-101 | x | 1.6 | 3.5 | 2.2 | 76.6 | 1.1 | 4.2 | 2.5 | 70.5 | 2.4 | 2.8 | 2.2 | 28.1 | 0.0 | 0.1 | 0.0 | 76.6 | 0.0 | 0.1 | 0.1 | 70.5 | 0.0 | 0.0 | 0.1 | 24.0 |
| | | DenseNet-161 | xi | 1.4 | 4.1 | 2.0 | 77.0 | 1.2 | 3.8 | 1.9 | 72.2 | 2.5 | 3.5 | 2.7 | 25.9 | 0.1 | 0.1 | 0.0 | 77.0 | 0.0 | 0.0 | 0.0 | 72.2 | 0.1 | 0.1 | 0.1 | 22.3 |
| | | AlexNet | xii | 2.9 | 4.1 | 2.9 | 76.5 | 2.5 | 4.2 | 2.2 | 70.0 | 2.0 | 3.4 | 2.0 | 27.8 | 0.0 | 0.1 | 0.1 | 76.4 | 0.0 | 0.3 | 0.2 | 70.0 | 0.0 | 0.0 | 0.1 | 24.2 |
| | image | Inception-v3 | xiii | 99.8 | 96.4 | 3.0 | 93 | 99.4 | 94.7 | 2.7 | 2.9 | 46.0 | 46.8 | 2.4 | 2.6 | 93.0 | 75.1 | 0.1 | 0.5 | 98.8 | 87.0 | 0.0 | 0.0 | 13.2 | 11.7 | 0.0 | 0.0 |
| | | ResNet-101 | xiv | 99.3 | 75.0 | 2.4 | 11.0 | 98.5 | 68.5 | 3.5 | 3.8 | 34.1 | 14.7 | 2.5 | 3.7 | 90.5 | 56.4 | 0.0 | 2.0 | 96.2 | 52.4 | 0.0 | 0.1 | 9.8 | 0.2 | 0.0 | 0.0 |
| | | DenseNet-161 | xv | 99.1 | 95.8 | 2.5 | 5.7 | 97.7 | 86.8 | 2.9 | 3.8 | 43.9 | 50.9 | 2.1 | 2.6 | 89.5 | 71.1 | 0.1 | 0.2 | 96.4 | 67.2 | 0.4 | 0.2 | 15.0 | 17.9 | 0.0 | 0.0 |
| | | AlexNet | xvi | 95.5 | 60.9 | 2.5 | 3.5 | 85.7 | 50.9 | 2.4 | 3.2 | 34.6 | 15.6 | 2.5 | 2.3 | 77.9 | 25.4 | 0.0 | 2.6 | 72.2 | 19.7 | 0.0 | 2.5 | 12.9 | 4.2 | 0.0 | 2.3 |
| Li et al. (2021) randomized | EEG | Inception-v3 | xvii | 2.6 | 2.5 | 2.5 | 1.6 | 2.6 | 2.9 | 2.6 | 2.5 | 3.0 | 2.2 | 2.7 | 1.6 | 0.1 | 0.1 | 0.1 | 1.5 | 0.1 | 0.1 | 0.0 | 2.5 | 0.2 | 0.1 | 0.2 | 1.4 |
| | | ResNet-101 | xviii | 2.6 | 2.4 | 2.8 | 2.1 | 2.8 | 2.7 | 2.9 | 2.3 | 2.6 | 2.7 | 3.0 | 1.8 | 0.0 | 0.1 | 0.1 | 1.9 | 0.0 | 0.1 | 0.2 | 2.3 | 0.1 | 0.2 | 0.1 | 1.6 |
| | | DenseNet-161 | xix | 2.1 | 2.2 | 2.3 | 1.5 | 2.5 | 2.3 | 2.5 | 2.0 | 3.1 | 2.0 | 1.9 | 1.9 | 0.1 | 0.1 | 0.1 | 1.5 | 0.0 | 0.0 | 0.1 | 2.0 | 0.1 | 0.1 | 0.1 | 1.7 |
| | | AlexNet | xx | 2.5 | 2.8 | 3.0 | 2.2 | 2.2 | 2.4 | 2.9 | 1.6 | 2.6 | 2.8 | 3.1 | 2.4 | 0.2 | 0.1 | 0.2 | 2.2 | 0.3 | 0.0 | 0.1 | 1.6 | 0.1 | 0.1 | 0.1 | 1.9 |
| | image | Inception-v3 | xxi | 99.8 | 91.1 | 3.4 | 9.4 | 99.4 | 84.7 | 3.0 | 2.9 | 46.0 | 57.1 | 2.1 | 2.5 | 93.0 | 64.0 | 0.0 | 0.6 | 98.8 | 73.3 | 0.0 | 0.0 | 13.2 | 20.4 | 0.0 | 0.0 |
| | | ResNet-101 | xxii | 99.3 | 69.8 | 2.9 | 11.2 | 98.5 | 53.1 | 2.6 | 4.0 | 34.1 | 9.4 | 3.1 | 2.2 | 90.5 | 42.0 | 0.0 | 2.0 | 96.2 | 28.5 | 0.0 | 0.0 | 9.8 | 3.5 | 0.0 | 0.0 |
| | | DenseNet-161 | xxiii | 99.1 | 65.6 | 2.1 | 5.1 | 97.7 | 36.0 | 2.5 | 3.0 | 43.9 | 36.1 | 2.3 | 3.0 | 89.5 | 30.5 | 0.1 | 3.0 | 96.4 | 11.7 | 0.0 | 0.0 | 15.0 | 13.6 | 0.0 | 0.0 |
| | | AlexNet | xxiv | 95.5 | 3.6 | 2.4 | 3.5 | 85.7 | 3.6 | 2.5 | 3.5 | 34.6 | 2.5 | 2.5 | 2.5 | 77.9 | 0.0 | 0.0 | 2.6 | 72.2 | 0.0 | 0.0 | 2.6 | 12.9 | 0.0 | 0.0 | 2.5 |

Table 16: The image encoder can generalize in some situations (some of columns ii, iv, vi, viii, x, xii, xiv, xvi, xviii, xx, xxii, and xxiv, for rows v–viii, xiii–xvi, and xxi–xxiv, are above chance). This, together with Tables 17 and 18, supports claim D.

| | | | original | | | | 40 classes top 40 | | | | bottom 960 | | | | original | | | | 1000 classes top 40 | | | | bottom 960 | | | |
|---|---|---|---|---|---|---|---|---|---|---|---|---|---|---|---|---|---|---|---|---|---|---|---|---|---|
| | | | before | after joint pretr. | after joint no pretr. | after separate | before | after joint pretr. | after joint no pretr. | after separate | before | after joint pretr. | after joint no pretr. | after separate | before | after joint pretr. | after joint no pretr. | after separate | before | after joint pretr. | after joint no pretr. | after separate | before | after joint pretr. | after joint no pretr. | after separate |
| | | | i | ii | iii | iv | v | vi | vii | viii | ix | x | xi | xii | xiii | xiv | xv | xvi | xvii | xviii | xix | xx | xxi | xxii | xxiii | xxiv |
| Spampinato et al. (2017) | EEG | Inception v3 | 2.6 | 3.0 | 2.7 | 55.5 | 2.2 | 2.6 | 2.5 | 55.5 | 2.5 | 3.1 | 2.5 | 2.3 | 0.1 | 0.1 | 0.1 | 55.5 | 0.1 | 0.1 | 0.1 | 55.5 | 0.1 | 0.1 | 0.1 | 2.3 |
| | | ResNet-101 | 2.3 | 3.0 | 2.4 | 54.7 | 2.8 | 2.4 | 2.4 | 54.7 | 2.4 | 3.1 | 2.4 | 2.3 | 0.0 | 0.0 | 0.1 | 54.7 | 0.0 | 0.0 | 0.1 | 54.7 | 0.1 | 0.0 | 0.1 | 2.3 |
| | | DenseNet-161 | 2.5 | 3.1 | 2.4 | 57.4 | 2.4 | 2.6 | 2.3 | 57.4 | 2.5 | 3.2 | 2.4 | 2.5 | 0.1 | 0.1 | 0.1 | 57.4 | 0.0 | 0.1 | 0.0 | 57.4 | 0.1 | 0.1 | 0.1 | 2.5 |
| | | AlexNet | 2.3 | 2.5 | 2.4 | 57.3 | 2.4 | 2.6 | 2.6 | 57.3 | 2.4 | 2.5 | 2.6 | 2.5 | 0.1 | 0.1 | 0.1 | 57.3 | 0.0 | 0.1 | 0.1 | 57.3 | 0.1 | 0.1 | 0.1 | 2.5 |
| | image | Inception v3 | 99.7 | 43.2 | 2.7 | 3.4 | 99.4 | 13.5 | 2.9 | 2.2 | 45.5 | 29.4 | 2.7 | 2.5 | 92.8 | 16.5 | 0.0 | 0.0 | 98.9 | 4.0 | 0.0 | 0.0 | 12.8 | 9.3 | 0.0 | 0.0 |
| | | ResNet-101 | 99.4 | 24.2 | 2.4 | 82.5 | 98.3 | 7.3 | 2.9 | 70.5 | 33.0 | 9.5 | 2.1 | 39.4 | 90.5 | 2.6 | 0.0 | 82.5 | 95.6 | 0.3 | 0.0 | 70.5 | 9.5 | 0.7 | 0.0 | 39.2 |
| | | DenseNet-161 | 99.2 | 7.2 | 2.4 | 23.3 | 97.9 | 4.4 | 2.3 | 3.5 | 42.7 | 3.8 | 2.6 | 2.6 | 89.4 | 0.1 | 0.0 | 12.1 | 95.8 | 0.0 | 0.0 | 0.1 | 14.2 | 0.0 | 0.1 | 0.0 |
| | | AlexNet | 95.8 | 2.6 | 2.7 | 18.7 | 85.8 | 2.6 | 2.7 | 19.1 | 34.1 | 2.6 | 2.4 | 2.4 | 78.4 | 0.0 | 0.3 | 18.7 | 72.5 | 0.0 | 0.3 | 19.1 | 12.5 | 0.0 | 0.0 | 2.4 |
| Li et al. (2021) Block | EEG | Inception v3 | 2.4 | 6.8 | 1.9 | 77.0 | 1.8 | 7.1 | 2.0 | 69.8 | 2.6 | 3.6 | 2.5 | 32.1 | 0.1 | 0.4 | 0.0 | 77.0 | 0.0 | 0.2 | 0.1 | 69.8 | 0.0 | 0.2 | 0.1 | 28.0 |
| | | ResNet-101 | 1.6 | 3.5 | 2.2 | 76.6 | 1.1 | 4.2 | 2.5 | 70.5 | 2.4 | 2.8 | 2.2 | 28.1 | 0.0 | 0.1 | 0.1 | 76.6 | 0.0 | 0.1 | 0.1 | 70.5 | 0.0 | 0.0 | 0.1 | 24.0 |
| | | DenseNet-161 | 1.4 | 4.1 | 2.0 | 77.0 | 1.2 | 3.8 | 1.9 | 72.2 | 2.5 | 3.5 | 2.7 | 25.9 | 0.1 | 0.1 | 0.0 | 77.0 | 0.0 | 0.0 | 0.0 | 72.2 | 0.1 | 0.1 | 0.1 | 22.3 |
| | | AlexNet | 2.9 | 4.1 | 2.9 | 76.5 | 2.5 | 4.2 | 2.2 | 70.0 | 2.0 | 3.4 | 2.5 | 27.8 | 0.0 | 0.1 | 0.1 | 76.4 | 0.0 | 0.3 | 0.2 | 70.0 | 0.0 | 0.0 | 0.2 | 24.2 |
| | image | Inception v3 | 99.8 | 96.4 | 3.0 | 9.3 | 99.4 | 94.7 | 3.1 | 2.9 | 46.0 | 46.8 | 2.4 | 2.5 | 93.0 | 75.1 | 0.1 | 0.5 | 98.8 | 87.0 | 0.0 | 0.0 | 13.2 | 11.7 | 0.0 | 0.0 |
| | | ResNet-101 | 99.3 | 75.0 | 2.4 | 11.0 | 98.5 | 68.5 | 2.7 | 3.5 | 34.1 | 14.7 | 2.5 | 3.7 | 90.5 | 56.4 | 0.0 | 2.0 | 96.2 | 52.4 | 0.0 | 0.1 | 9.8 | 0.2 | 0.0 | 0.0 |
| | | DenseNet-161 | 99.1 | 95.8 | 2.5 | 5.7 | 97.7 | 86.8 | 2.9 | 3.8 | 43.9 | 50.9 | 2.1 | 2.6 | 89.5 | 71.1 | 0.1 | 0.2 | 96.4 | 67.2 | 0.4 | 0.2 | 15.0 | 17.9 | 0.0 | 0.0 |
| | | AlexNet | 95.5 | 60.9 | 2.5 | 3.5 | 85.7 | 50.9 | 2.4 | 3.2 | 34.6 | 15.6 | 2.5 | 2.3 | 77.9 | 25.4 | 0.0 | 2.6 | 72.2 | 19.7 | 0.0 | 2.5 | 12.9 | 4.2 | 0.0 | 2.3 |
| Li et al. (2021) randomized | EEG | Inception v3 | 2.6 | 2.5 | 2.5 | 1.6 | 2.6 | 2.9 | 2.6 | 2.5 | 3.0 | 2.2 | 2.7 | 1.6 | 0.0 | 0.1 | 0.1 | 1.5 | 0.1 | 0.1 | 0.0 | 2.5 | 0.2 | 0.1 | 0.2 | 1.4 |
| | | ResNet-101 | 2.6 | 2.4 | 2.8 | 2.1 | 2.8 | 2.7 | 2.9 | 2.3 | 2.6 | 2.7 | 3.0 | 1.8 | 0.0 | 0.1 | 0.1 | 1.9 | 0.0 | 0.1 | 0.2 | 2.3 | 0.1 | 0.2 | 0.1 | 1.6 |
| | | DenseNet-161 | 2.1 | 2.2 | 2.3 | 1.5 | 2.5 | 2.3 | 2.5 | 2.0 | 3.1 | 2.0 | 1.9 | 1.9 | 0.1 | 0.1 | 0.1 | 1.5 | 0.0 | 0.0 | 0.1 | 2.0 | 0.1 | 0.1 | 0.1 | 1.7 |
| | | AlexNet | 2.5 | 2.8 | 3.0 | 2.2 | 2.2 | 2.4 | 2.9 | 1.6 | 2.6 | 2.8 | 3.1 | 2.4 | 0.2 | 0.1 | 0.2 | 2.2 | 0.3 | 0.0 | 0.1 | 1.6 | 0.1 | 0.1 | 0.1 | 1.9 |
| | image | Inception v3 | 99.8 | 91.1 | 3.4 | 9.4 | 99.4 | 84.7 | 3.0 | 2.9 | 46.0 | 57.1 | 2.1 | 2.5 | 93.0 | 64.0 | 0.1 | 0.6 | 98.8 | 73.3 | 0.0 | 0.0 | 13.2 | 20.4 | 0.0 | 0.0 |
| | | ResNet-101 | 99.3 | 69.8 | 2.9 | 1.2 | 98.5 | 53.1 | 2.6 | 4.0 | 34.1 | 9.4 | 3.1 | 2.2 | 90.5 | 42.0 | 0.0 | 2.0 | 96.2 | 28.5 | 0.2 | 0.1 | 9.8 | 3.5 | 0.0 | 0.0 |
| | | DenseNet-161 | 99.1 | 65.6 | 2.1 | 5.1 | 97.7 | 36.0 | 2.3 | 3.0 | 43.9 | 36.1 | 2.0 | 3.0 | 89.5 | 30.5 | 0.1 | 0.0 | 96.4 | 11.7 | 0.1 | 0.0 | 15.0 | 13.6 | 0.0 | 0.0 |
| | | AlexNet | 95.5 | 3.6 | 2.4 | 3.5 | 85.7 | 3.6 | 2.5 | 3.5 | 34.6 | 2.5 | 2.5 | 2.5 | 77.9 | 0.0 | 0.0 | 2.6 | 72.2 | 0.0 | 0.0 | 2.6 | 12.9 | 0.0 | 0.0 | 2.5 |

Table 17: The EEG encoder can generalize in some situations on confounded data with separate training (some of columns iv, viii, xii, xvi, xx, and xxiv, for rows i–iv and ix–xii, are above chance). This, together with Tables 16 and 18, supports claim D.

| | | | original after joint | | | 40 classes top 40 after joint | | | | bottom 960 after joint | | | | original after joint | | | | 1000 classes top 40 after joint | | | | bottom 960 after joint | | | |
| | | before | pret. | no pret. | after sep. | before | pret. | no pret. | after sep. | before | pret. | no pret. | after sep. | before | pret. | no pret. | after sep. | before | pret. | no pret. | after sep. | before | pret. | no pret. | after sep. |
| | | i | ii | iii | iv | v | vi | vii | viii | ix | x | xi | xii | xiii | xiv | xv | xvi | xvii | xviii | xix | xx | xxi | xxii | xxiii | xxiv |
|---|---|---|---|---|---|---|---|---|---|---|---|---|---|---|---|---|---|---|---|---|---|---|---|---|---|
| Spampinato et al. (2017) EEG | Inception v3 | 2.6 | 3.0 | 2.7 | 55.5 | 2.2 | 2.6 | 2.5 | 55.5 | 2.5 | 3.1 | 2.5 | 2.3 | 0.1 | 0.1 | 0.1 | 55.5 | 0.1 | 0.1 | 0.1 | 55.5 | 0.1 | 0.1 | 0.1 | 2.3 |
| | ResNet-101 | 2.3 | 3.0 | 2.4 | 54.7 | 2.8 | 2.4 | 2.4 | 54.7 | 2.4 | 3.1 | 2.4 | 2.3 | 0.0 | 0.1 | 0.1 | 54.7 | 0.0 | 0.0 | 0.0 | 54.7 | 0.1 | 0.1 | 0.1 | 2.3 |
| | DenseNet-161 | 2.5 | 3.1 | 2.4 | 57.4 | 2.4 | 2.6 | 2.3 | 57.4 | 2.5 | 3.2 | 2.4 | 2.5 | 0.1 | 0.1 | 0.0 | 57.4 | 0.0 | 0.1 | 0.0 | 57.4 | 0.1 | 0.1 | 0.1 | 2.5 |
| | AlexNet | 2.3 | 2.5 | 2.4 | 57.3 | 2.4 | 2.6 | 2.6 | 57.3 | 2.4 | 2.5 | 2.6 | 2.5 | 0.1 | 0.1 | 0.1 | 57.3 | 0.0 | 0.1 | 0.1 | 57.3 | 0.1 | 0.1 | 0.1 | 2.5 |
| image | Inception v3 | 99.7 | 43.2 | 2.7 | 3.4 | 99.4 | 13.5 | 2.9 | 2.2 | 45.5 | 29.4 | 2.7 | 2.5 | 92.8 | 16.5 | 0.0 | 0.0 | 98.9 | 4.0 | 0.0 | 0.0 | 12.8 | 9.3 | 0.0 | 0.0 |
| | ResNet-101 | 99.4 | 24.2 | 2.4 | 82.5 | 98.3 | 7.3 | 2.9 | 70.5 | 33.0 | 9.5 | 2.1 | 39.4 | 90.5 | 2.6 | 0.0 | 82.5 | 95.6 | 0.3 | 0.0 | 70.5 | 9.5 | 0.7 | 0.0 | 39.2 |
| | DenseNet-161 | 99.2 | 7.2 | 2.4 | 23.3 | 97.9 | 4.4 | 2.3 | 3.5 | 42.7 | 3.8 | 2.6 | 2.6 | 89.4 | 0.1 | 0.0 | 12.1 | 95.8 | 0.0 | 0.0 | 0.1 | 14.2 | 0.0 | 0.1 | 0.0 |
| | AlexNet | 95.8 | 2.6 | 2.7 | 18.7 | 85.8 | 2.6 | 2.7 | 19.1 | 34.1 | 2.6 | 2.4 | 2.4 | 78.4 | 0.0 | 0.3 | 18.7 | 72.5 | 0.0 | 0.3 | 19.1 | 12.5 | 0.0 | 0.0 | 2.4 |
| Li et al. (2021) Block EEG | Inception v3 | 2.4 | 6.8 | 1.9 | 77.0 | 1.8 | 7.1 | 2.0 | 69.8 | 2.6 | 3.6 | 2.5 | 32.1 | 0.1 | 0.4 | 0.0 | 77.0 | 0.0 | 0.2 | 0.1 | 69.8 | 0.0 | 0.2 | 0.1 | 28.0 |
| | ResNet-101 | 1.6 | 3.5 | 2.2 | 76.6 | 1.1 | 4.2 | 2.5 | 70.5 | 2.4 | 2.8 | 2.2 | 28.1 | 0.0 | 0.1 | 0.1 | 76.6 | 0.0 | 0.1 | 0.1 | 70.5 | 0.0 | 0.0 | 0.1 | 24.0 |
| | DenseNet-161 | 1.4 | 4.1 | 2.0 | 77.0 | 1.2 | 3.8 | 1.9 | 72.2 | 2.5 | 3.5 | 2.7 | 25.9 | 0.1 | 0.1 | 0.0 | 77.0 | 0.0 | 0.0 | 0.0 | 72.2 | 0.1 | 0.1 | 0.1 | 22.3 |
| | AlexNet | 2.9 | 4.1 | 2.9 | 76.5 | 2.5 | 4.2 | 2.2 | 70.0 | 2.0 | 3.4 | 2.5 | 27.8 | 0.0 | 0.1 | 0.1 | 76.4 | 0.0 | 0.3 | 0.2 | 70.0 | 0.0 | 0.0 | 0.1 | 24.2 |
| image | Inception v3 | 99.8 | 96.4 | 3.0 | 9.3 | 99.4 | 94.7 | 3.1 | 2.9 | 46.0 | 46.8 | 2.4 | 2.5 | 93.0 | 75.1 | 0.1 | 0.5 | 98.8 | 87.0 | 0.0 | 0.0 | 13.2 | 11.7 | 0.0 | 0.0 |
| | ResNet-101 | 99.3 | 75.0 | 2.4 | 11.0 | 98.5 | 68.5 | 2.7 | 3.5 | 34.1 | 14.7 | 2.5 | 3.7 | 90.5 | 56.4 | 0.0 | 2.0 | 96.2 | 52.4 | 0.2 | 0.1 | 9.8 | 0.2 | 0.0 | 0.0 |
| | DenseNet-161 | 99.1 | 95.8 | 2.5 | 5.7 | 97.7 | 86.8 | 2.9 | 3.8 | 43.9 | 50.9 | 2.1 | 2.6 | 89.5 | 71.1 | 0.1 | 0.2 | 96.4 | 67.2 | 0.4 | 0.2 | 15.0 | 17.9 | 0.0 | 0.0 |
| | AlexNet | 95.5 | 60.9 | 2.5 | 3.5 | 85.7 | 50.9 | 2.4 | 3.2 | 34.6 | 15.6 | 2.5 | 2.3 | 77.9 | 25.4 | 0.0 | 2.6 | 72.2 | 19.7 | 0.0 | 2.5 | 12.9 | 4.2 | 0.0 | 2.3 |
| Li et al. (2021) randomized EEG | Inception v3 | 2.6 | 2.5 | 2.5 | 1.6 | 2.6 | 2.9 | 2.6 | 2.5 | 3.0 | 2.2 | 2.7 | 1.6 | 0.1 | 0.1 | 0.1 | 1.5 | 0.1 | 0.1 | 0.2 | 2.5 | 0.1 | 0.1 | 0.0 | 1.4 |
| | ResNet-101 | 2.6 | 2.4 | 2.8 | 2.1 | 2.8 | 2.7 | 2.9 | 2.3 | 2.6 | 2.7 | 3.0 | 1.8 | 0.0 | 0.1 | 0.1 | 1.9 | 0.0 | 0.1 | 0.2 | 2.3 | 0.1 | 0.2 | 0.1 | 1.6 |
| | DenseNet-161 | 2.1 | 2.2 | 2.3 | 1.5 | 2.5 | 2.3 | 2.5 | 2.0 | 3.1 | 2.0 | 1.9 | 1.9 | 0.1 | 0.1 | 0.1 | 1.5 | 0.0 | 0.0 | 0.1 | 2.0 | 0.1 | 0.1 | 0.1 | 1.7 |
| | AlexNet | 2.5 | 2.8 | 3.0 | 2.2 | 2.2 | 2.4 | 2.9 | 1.6 | 2.6 | 2.8 | 3.1 | 2.4 | 0.2 | 0.1 | 0.2 | 2.2 | 0.3 | 0.0 | 0.1 | 1.6 | 0.1 | 0.1 | 0.1 | 1.9 |
| image | Inception v3 | 99.8 | 91.1 | 3.4 | 9.4 | 99.4 | 84.7 | 3.0 | 2.9 | 46.0 | 57.1 | 3.1 | 2.5 | 93.0 | 64.0 | 0.1 | 0.6 | 98.8 | 73.3 | 0.0 | 0.0 | 13.2 | 20.4 | 0.0 | 0.0 |
| | ResNet-101 | 99.3 | 69.8 | 2.9 | 11.2 | 98.5 | 53.1 | 2.6 | 4.0 | 34.1 | 9.4 | 3.1 | 2.2 | 90.5 | 42.0 | 0.0 | 2.0 | 96.2 | 28.5 | 0.0 | 0.1 | 9.8 | 3.5 | 0.0 | 0.0 |
| | DenseNet-161 | 99.1 | 65.6 | 2.1 | 5.1 | 97.7 | 36.0 | 2.5 | 3.0 | 43.9 | 36.1 | 2.3 | 3.0 | 89.5 | 30.5 | 0.1 | 0.0 | 96.4 | 11.7 | 0.0 | 0.0 | 15.0 | 13.6 | 0.0 | 0.0 |
| | AlexNet | 95.5 | 3.6 | 2.4 | 3.5 | 85.7 | 3.6 | 2.5 | 3.5 | 34.6 | 2.5 | 2.5 | 2.5 | 77.9 | 0.0 | 0.0 | 2.6 | 72.2 | 0.0 | 0.0 | 2.6 | 12.9 | 0.0 | 0.0 | 2.5 |

Table 18: The EEG encoder cannot generalize on nonconfounded data with joint training with pretraining (all of columns ii, vi, x, xiv, xviii, and xxi, for rows xvii–xx, are at chance). This, together with Tables 16 and 17, supports claim D.

The table is organized into two macro-groups — **40 classes** (columns i–xii) and **1000 classes** (columns xiii–xxiv) — each subdivided into *original after joint* (before | pretraining | no pretraining | after separate), *top 40*, and *bottom 960*.

| Study | Mod. | Model | # | original after joint — before (i) | pretraining (ii) | no pretraining (iii) | after separate (iv) | top 40 — before (v) | pretraining (vi) | no pretraining (vii) | after separate (viii) | bottom 960 — before (ix) | pretraining (x) | no pretraining (xi) | after separate (xii) | original after joint — before (xiii) | pretraining (xiv) | no pretraining (xv) | after separate (xvi) | top 40 — before (xvii) | pretraining (xviii) | no pretraining (xix) | after separate (xx) | bottom 960 — before (xxi) | pretraining (xxii) | no pretraining (xxiii) | after separate (xxiv) |
|---|---|---|---|---|---|---|---|---|---|---|---|---|---|---|---|---|---|---|---|---|---|---|---|---|---|---|---|
| Spampinato et al. (2017) | EEG | Inception v3 | i | 2.6 | 3.0 | 2.7 | 55.5 | 2.2 | 2.6 | 2.5 | 55.5 | 2.5 | 3.1 | 2.5 | 2.3 | 0.1 | 0.1 | 0.1 | 55.5 | 0.1 | 0.1 | 0.1 | 55.5 | 0.1 | 0.1 | 0.1 | 2.3 |
| | | ResNet-101 | ii | 2.3 | 3.0 | 2.4 | 54.7 | 2.8 | 2.4 | 2.4 | 54.7 | 2.4 | 3.1 | 2.5 | 2.3 | 0.0 | 0.1 | 0.1 | 54.7 | 0.0 | 0.0 | 0.1 | 54.7 | 0.1 | 0.1 | 0.1 | 2.3 |
| | | DenseNet-161 | iii | 2.5 | 3.1 | 2.4 | 57.4 | 2.8 | 2.6 | 2.3 | 57.4 | 2.5 | 3.2 | 2.4 | 2.5 | 0.1 | 0.1 | 0.1 | 57.4 | 0.1 | 0.1 | 0.0 | 57.4 | 0.1 | 0.1 | 0.1 | 2.5 |
| | | AlexNet | iv | 2.3 | 2.5 | 2.4 | 57.3 | 2.4 | 2.6 | 2.6 | 57.3 | 2.4 | 2.5 | 2.6 | 2.5 | 0.1 | 0.1 | 0.1 | 57.3 | 0.0 | 0.1 | 0.1 | 57.3 | 0.1 | 0.1 | 0.1 | 2.5 |
| | image | Inception v3 | v | 99.7 | 43.2 | 2.7 | 3.4 | 99.4 | 13.5 | 2.9 | 2.2 | 45.5 | 29.4 | 2.7 | 2.5 | 92.8 | 16.5 | 0.0 | 0.0 | 98.9 | 4.0 | 0.0 | 0.0 | 12.8 | 9.3 | 0.0 | 0.0 |
| | | ResNet-101 | vi | 99.4 | 24.2 | 2.4 | 82.5 | 98.3 | 7.3 | 2.9 | 70.5 | 33.0 | 9.5 | 2.1 | 39.4 | 90.5 | 2.6 | 0.0 | 82.5 | 95.6 | 0.3 | 0.0 | 70.5 | 9.5 | 0.7 | 0.0 | 39.2 |
| | | DenseNet-161 | vii | 99.2 | 7.2 | 2.4 | 23.3 | 97.9 | 4.4 | 2.3 | 3.5 | 42.7 | 3.8 | 2.6 | 2.6 | 89.4 | 0.1 | 0.0 | 12.1 | 95.8 | 0.0 | 0.0 | 0.1 | 14.2 | 0.0 | 0.1 | 0.0 |
| | | AlexNet | viii | 95.8 | 2.6 | 2.7 | 18.7 | 85.8 | 2.6 | 2.7 | 19.1 | 34.1 | 2.6 | 2.4 | 2.4 | 78.4 | 0.0 | 0.3 | 18.7 | 72.5 | 0.0 | 0.3 | 19.1 | 12.5 | 0.0 | 0.0 | 2.4 |
| Li et al. (2021) Block | EEG | Inception v3 | ix | 2.4 | 6.8 | 1.9 | 77.0 | 1.8 | 7.1 | 2.0 | 69.8 | 2.6 | 3.6 | 2.5 | 32.1 | 0.1 | 0.4 | 0.0 | 77.0 | 0.0 | 0.2 | 0.1 | 69.8 | 0.0 | 0.2 | 0.1 | 28.0 |
| | | ResNet-101 | x | 1.6 | 3.5 | 2.2 | 76.6 | 1.1 | 4.2 | 2.5 | 70.5 | 2.4 | 2.8 | 2.2 | 28.1 | 0.0 | 0.1 | 0.1 | 76.6 | 0.0 | 0.1 | 0.0 | 70.5 | 0.0 | 0.0 | 0.1 | 24.0 |
| | | DenseNet-161 | xi | 1.4 | 4.1 | 2.0 | 77.0 | 1.2 | 3.8 | 1.9 | 72.2 | 2.5 | 3.5 | 2.7 | 25.9 | 0.1 | 0.1 | 0.0 | 77.0 | 0.0 | 0.0 | 0.0 | 72.2 | 0.1 | 0.1 | 0.1 | 22.3 |
| | | AlexNet | xii | 2.9 | 4.1 | 2.9 | 76.5 | 2.5 | 4.2 | 2.2 | 70.0 | 2.0 | 3.4 | 2.5 | 27.8 | 0.0 | 0.1 | 0.1 | 76.4 | 0.0 | 0.3 | 0.2 | 70.0 | 0.0 | 0.0 | 0.1 | 24.2 |
| | image | Inception v3 | xiii | 99.8 | 96.4 | 3.0 | 9.3 | 99.4 | 94.7 | 3.1 | 2.9 | 46.0 | 46.8 | 2.4 | 2.5 | 93.0 | 75.1 | 0.1 | 0.5 | 98.8 | 87.0 | 0.0 | 0.0 | 13.2 | 11.7 | 0.0 | 0.0 |
| | | ResNet-101 | xiv | 99.3 | 75.0 | 2.4 | 11.0 | 98.5 | 68.5 | 2.7 | 3.5 | 34.1 | 14.7 | 2.5 | 3.7 | 90.5 | 56.4 | 0.1 | 2.0 | 96.2 | 52.4 | 0.1 | 0.1 | 9.8 | 0.2 | 0.1 | 1.6 |
| | | DenseNet-161 | xv | 99.1 | 95.8 | 2.5 | 5.7 | 97.7 | 86.8 | 2.9 | 3.8 | 43.9 | 50.9 | 2.1 | 2.6 | 89.5 | 71.1 | 0.1 | 0.2 | 96.4 | 67.2 | 0.0 | 0.0 | 15.0 | 17.9 | 0.0 | 1.7 |
| | | AlexNet | xvi | 95.5 | 60.9 | 2.5 | 3.5 | 85.7 | 50.9 | 2.4 | 3.2 | 34.6 | 15.6 | 2.5 | 2.3 | 77.9 | 25.4 | 0.0 | 2.6 | 72.2 | 19.7 | 0.0 | 2.5 | 12.9 | 4.2 | 0.0 | 2.3 |
| Li et al. (2021) randomized | EEG | Inception v3 | xvii | 2.6 | 2.5 | 2.5 | 1.6 | 2.6 | 2.9 | 2.6 | 2.5 | 3.0 | 2.2 | 2.7 | 1.6 | 0.1 | 0.1 | 0.1 | 1.5 | 0.1 | 0.1 | 0.0 | 2.5 | 0.2 | 0.1 | 0.1 | 1.4 |
| | | ResNet-101 | xviii | 2.6 | 2.2 | 2.8 | 2.1 | 2.8 | 2.7 | 2.9 | 2.3 | 2.6 | 2.7 | 3.0 | 1.8 | 0.1 | 0.1 | 0.1 | 1.9 | 0.0 | 0.1 | 0.2 | 2.3 | 0.1 | 0.2 | 0.1 | 1.6 |
| | | DenseNet-161 | xix | 2.1 | 2.2 | 2.3 | 1.5 | 2.5 | 2.3 | 2.5 | 2.0 | 3.1 | 2.0 | 1.9 | 1.9 | 0.1 | 0.1 | 0.1 | 1.5 | 0.0 | 0.0 | 0.1 | 2.0 | 0.1 | 0.1 | 0.1 | 1.7 |
| | | AlexNet | xx | 2.5 | 2.8 | 3.0 | 2.2 | 2.2 | 2.4 | 2.9 | 1.6 | 2.6 | 2.8 | 3.1 | 2.4 | 0.2 | 0.1 | 0.2 | 2.2 | 0.3 | 0.0 | 0.1 | 1.6 | 0.1 | 0.1 | 0.1 | 1.9 |
| | image | Inception v3 | xxi | 99.8 | 91.1 | 3.4 | 9.4 | 99.4 | 84.7 | 3.0 | 2.9 | 46.0 | 57.1 | 2.1 | 2.5 | 93.0 | 64.0 | 0.1 | 0.6 | 98.8 | 73.3 | 0.1 | 0.0 | 13.2 | 20.4 | 0.0 | 0.0 |
| | | ResNet-101 | xxii | 99.3 | 69.8 | 2.9 | 11.2 | 98.5 | 53.1 | 2.6 | 4.0 | 34.1 | 9.4 | 3.1 | 2.2 | 90.5 | 42.0 | 0.1 | 2.0 | 96.2 | 28.5 | 0.0 | 3.5 | 9.8 | 3.5 | 0.0 | 0.0 |
| | | DenseNet-161 | xxiii | 99.1 | 65.6 | 2.1 | 5.1 | 97.7 | 36.0 | 2.5 | 3.0 | 43.9 | 36.1 | 2.3 | 3.0 | 89.5 | 30.5 | 0.1 | 0.0 | 96.4 | 11.7 | 0.0 | 0.0 | 15.0 | 13.6 | 0.0 | 0.0 |
| | | AlexNet | xxiv | 95.5 | 3.6 | 2.4 | 3.5 | 85.7 | 3.6 | 2.5 | 3.5 | 34.6 | 2.5 | 2.5 | 2.5 | 77.9 | 0.0 | 0.0 | 2.6 | 72.2 | 0.0 | 0.0 | 2.6 | 12.9 | 0.0 | 0.0 | 2.5 |

Table 19: Classification accuracy is at chance after joint training without pretraining (columns iii vii, xi, xv, xix, and xxiii). This holds both for EEG classification and image classification, both for 40 classes and 1000 classes, both for the 'Spampinato et al. (2017)' data, the 'Li et al. (2021) block' data, and the 'Li et al. (2021) randomized' data, and whether using all components, just the 40 principal components, or the bottom 960 components. This suggests that their method completely breaks down when not provided with class information through pretraining, and further supports claim A and calls claim I into question.

| | | | 40 classes | | | | | | | | | | | | 1000 classes | | | | | | | | | | | |
| | | | original | | | | top 40 | | | | bottom 960 | | | | original | | | | top 40 | | | | bottom 960 | | | |
| | | | before | after joint | | after separate | before | after joint | | after separate | before | after joint | | after separate | before | after joint | | after separate | before | after joint | | after separate | before | after joint | | after separate |
| Dataset | Type | Model | | pretraining | no pretraining | | | pretraining | no pretraining | | | pretraining | no pretraining | | | pretraining | no pretraining | | | pretraining | no pretraining | | | pretraining | no pretraining | |
| | | | i | ii | iii | iv | v | vi | vii | viii | ix | x | xi | xii | xiii | xiv | xv | xvi | xvii | xviii | xix | xx | xxi | xxii | xxiii | xxiv |
| Spampinato et al. (2017) | EEG | Inception v3 | 2.6 | 3.0 | 2.5 | 55.5 | 2.2 | 2.6 | 2.5 | 55.5 | 2.5 | 3.1 | 2.5 | 2.3 | 0.1 | 0.1 | 0.1 | 55.5 | 0.1 | 0.1 | 0.1 | 55.5 | 0.1 | 0.1 | 0.1 | 2.3 |
| | | ResNet-101 | 2.3 | 3.0 | 2.6 | 54.7 | 2.8 | 2.4 | 2.4 | 54.7 | 2.4 | 3.1 | 2.5 | 2.3 | 0.0 | 0.1 | 0.1 | 54.7 | 0.0 | 0.0 | 0.1 | 54.7 | 0.1 | 0.1 | 0.1 | 2.3 |
| | | DenseNet-161 | 2.5 | 3.1 | 2.4 | 57.4 | 2.4 | 2.6 | 2.3 | 57.4 | 2.5 | 3.2 | 2.4 | 2.5 | 0.1 | 0.1 | 0.1 | 57.4 | 0.0 | 0.1 | 0.0 | 57.4 | 0.1 | 0.1 | 0.1 | 2.5 |
| | | AlexNet | 2.3 | 2.5 | 2.4 | 57.3 | 2.4 | 2.6 | 2.6 | 57.3 | 2.4 | 2.5 | 2.6 | 2.5 | 0.1 | 0.1 | 0.1 | 57.3 | 0.0 | 0.1 | 0.1 | 57.3 | 0.1 | 0.1 | 0.1 | 2.5 |
| | image | Inception v3 | 99.7 | 43.2 | 2.7 | 3.4 | 99.4 | 13.5 | 2.9 | 2.2 | 45.5 | 29.4 | 2.7 | 2.5 | 92.8 | 16.5 | 0.0 | 0.0 | 98.9 | 4.0 | 0.0 | 0.0 | 12.8 | 9.3 | 0.0 | 0.0 |
| | | ResNet-101 | 99.4 | 24.2 | 2.4 | 82.5 | 98.3 | 7.3 | 2.9 | 70.5 | 33.0 | 9.5 | 2.1 | 39.4 | 90.5 | 2.6 | 0.0 | 82.5 | 95.6 | 0.3 | 0.0 | 70.5 | 9.5 | 0.7 | 0.0 | 39.2 |
| | | DenseNet-161 | 99.2 | 7.2 | 2.4 | 23.3 | 97.9 | 4.4 | 2.3 | 3.5 | 42.7 | 3.8 | 2.6 | 2.6 | 89.4 | 0.1 | 0.0 | 12.1 | 95.8 | 0.0 | 0.0 | 0.1 | 14.2 | 0.0 | 0.1 | 0.0 |
| | | AlexNet | 95.8 | 2.6 | 2.7 | 18.7 | 85.8 | 2.6 | 2.7 | 19.1 | 34.1 | 2.6 | 2.4 | 2.4 | 78.4 | 0.0 | 0.3 | 18.7 | 72.5 | 0.0 | 0.3 | 19.1 | 12.5 | 0.0 | 0.0 | 2.4 |
| Li et al. (2021) block | EEG | Inception v3 | 2.4 | 6.8 | 1.9 | 77.0 | 1.8 | 7.1 | 2.0 | 69.8 | 2.6 | 3.6 | 2.5 | 32.1 | 0.1 | 0.4 | 0.0 | 77.0 | 0.0 | 0.2 | 0.1 | 69.8 | 0.0 | 0.2 | 0.1 | 28.0 |
| | | ResNet-101 | 1.6 | 3.5 | 2.2 | 76.6 | 1.1 | 4.2 | 2.5 | 70.5 | 2.4 | 2.8 | 2.2 | 28.1 | 0.0 | 0.1 | 0.1 | 76.6 | 0.0 | 0.0 | 0.1 | 70.5 | 0.0 | 0.0 | 0.1 | 24.0 |
| | | DenseNet-161 | 1.4 | 4.1 | 2.0 | 77.0 | 1.2 | 3.8 | 1.9 | 72.2 | 2.5 | 3.5 | 2.7 | 25.9 | 0.1 | 0.1 | 0.1 | 77.0 | 0.0 | 0.0 | 0.2 | 72.2 | 0.1 | 0.1 | 0.1 | 22.3 |
| | | AlexNet | 2.9 | 4.1 | 2.9 | 76.5 | 2.5 | 4.2 | 2.2 | 70.0 | 2.0 | 3.4 | 2.5 | 27.8 | 0.0 | 0.1 | 0.1 | 76.4 | 0.0 | 0.3 | 0.2 | 70.0 | 0.0 | 0.1 | 0.1 | 24.2 |
| | image | Inception v3 | 99.8 | 96.4 | 3.0 | 9.3 | 99.4 | 94.7 | 3.1 | 9.3 | 46.0 | 46.8 | 2.4 | 2.5 | 93.0 | 75.1 | 0.1 | 0.5 | 98.8 | 87.0 | 0.0 | 0.0 | 13.2 | 11.7 | 0.0 | 0.0 |
| | | ResNet-101 | 99.3 | 75.0 | 2.4 | 11.0 | 98.5 | 68.5 | 2.7 | 3.5 | 34.1 | 14.7 | 2.5 | 3.7 | 90.5 | 56.4 | 0.1 | 2.0 | 96.2 | 52.4 | 0.0 | 0.1 | 9.8 | 0.2 | 0.0 | 0.0 |
| | | DenseNet-161 | 99.1 | 95.8 | 2.5 | 5.7 | 97.7 | 86.8 | 2.9 | 3.8 | 43.9 | 50.9 | 2.1 | 2.6 | 89.5 | 71.1 | 0.1 | 0.2 | 96.4 | 67.2 | 0.4 | 0.2 | 15.0 | 17.9 | 0.0 | 0.0 |
| | | AlexNet | 95.5 | 60.9 | 2.5 | 3.5 | 85.7 | 50.9 | 2.4 | 3.2 | 34.6 | 15.6 | 2.5 | 2.3 | 77.9 | 25.4 | 0.0 | 2.6 | 72.2 | 19.7 | 0.0 | 2.5 | 12.9 | 4.2 | 0.0 | 2.3 |
| Li et al. (2021) randomized | EEG | Inception v3 | 2.6 | 2.5 | 2.5 | 1.6 | 2.6 | 2.9 | 2.6 | 2.5 | 3.0 | 2.2 | 2.7 | 1.6 | 0.0 | 0.1 | 0.1 | 1.5 | 0.1 | 0.1 | 0.0 | 2.5 | 0.2 | 0.1 | 0.2 | 1.4 |
| | | ResNet-101 | 2.6 | 2.4 | 2.8 | 2.1 | 2.8 | 2.7 | 2.9 | 2.3 | 2.6 | 2.7 | 3.0 | 1.8 | 0.0 | 0.1 | 0.1 | 1.9 | 0.0 | 0.1 | 0.2 | 2.3 | 0.1 | 0.2 | 0.1 | 1.6 |
| | | DenseNet-161 | 2.1 | 2.2 | 2.3 | 1.5 | 2.5 | 2.3 | 2.5 | 2.0 | 3.1 | 2.0 | 1.9 | 1.9 | 0.1 | 0.1 | 0.2 | 1.5 | 0.0 | 0.0 | 0.1 | 2.0 | 0.1 | 0.1 | 0.1 | 1.7 |
| | | AlexNet | 2.5 | 2.8 | 3.0 | 2.2 | 2.2 | 2.4 | 2.9 | 1.6 | 2.6 | 2.8 | 3.1 | 2.4 | 0.2 | 0.1 | 0.1 | 2.2 | 0.3 | 0.0 | 0.1 | 1.6 | 0.1 | 0.1 | 0.1 | 1.9 |
| | image | Inception v3 | 99.8 | 91.1 | 3.4 | 9.4 | 99.4 | 84.7 | 3.0 | 2.9 | 46.0 | 57.1 | 2.1 | 2.5 | 93.0 | 64.0 | 0.1 | 0.6 | 98.8 | 73.3 | 0.0 | 0.0 | 13.2 | 20.4 | 0.0 | 0.0 |
| | | ResNet-101 | 99.3 | 69.8 | 2.9 | 11.2 | 98.5 | 53.1 | 2.6 | 4.0 | 34.1 | 9.4 | 3.1 | 2.2 | 90.5 | 42.0 | 0.0 | 2.0 | 96.2 | 28.5 | 0.0 | 0.1 | 9.8 | 3.5 | 0.0 | 0.0 |
| | | DenseNet-161 | 99.1 | 65.6 | 2.1 | 5.1 | 97.7 | 36.0 | 2.5 | 3.0 | 43.9 | 36.1 | 2.0 | 3.0 | 89.5 | 30.5 | 0.1 | 0.0 | 96.4 | 11.7 | 0.1 | 0.0 | 15.0 | 13.6 | 0.0 | 0.0 |
| | | AlexNet | 95.5 | 3.6 | 2.4 | 3.5 | 85.7 | 3.6 | 2.5 | 3.5 | 34.6 | 2.5 | 2.5 | 2.5 | 77.9 | 0.0 | 0.0 | 2.6 | 72.2 | 0.0 | 0.0 | 2.6 | 12.9 | 0.0 | 0.0 | 2.5 |

Table 20: With only two exceptions, separate training gets to a lower loss than joint training with pretraining (compare columns vii, viii, and ix to i, ii, and iii, respectively). This suggests that there is a point within the representational-capacity space of their model and loss function that achieves lower loss than achieved by their joint-training procedure. The joint-training procedure could have achieved it, it just didn't. Since that point was achieved with separate training, the resulting EEG encodings do not have any image information and the resulting image encodings do not have any brain-activity information. This supports claim B and calls claims II-IV into question.

| | | joint | | | no pretraining | | | separate | | |
| | | pretraining | | | | | | | | |
| | | Spampinato et al. (2017) | Li et al. (2021) block | Li et al. (2021) randomized | Spampinato et al. (2017) | Li et al. (2021) block | Li et al. (2021) randomized | Spampinato et al. (2017) | Li et al. (2021) block | Li et al. (2021) randomized |
| | | i | ii | iii | iv | v | vi | vii | viii | ix |
|---|---|---|---|---|---|---|---|---|---|---|
| Inception v3 | i | 1.303 | 1.098 | 0.932 | 1.074 | 0.938 | 0.919 | 1.150 | 1.027 | 1.104 |
| ResNet-101 | ii | 1.155 | 0.925 | 1.019 | 0.963 | 1.006 | 1.080 | 0.885 | 0.936 | 0.884 |
| DenseNet-161 | iii | 1.027 | 1.080 | 1.047 | 0.988 | 0.919 | 0.929 | 1.007 | 0.982 | 1.029 |
| AlexNet | iv | 1.417 | 1.142 | 0.981 | 1.136 | 1.055 | 1.034 | 0.925 | 0.956 | 0.935 |

Table 21: With only three exceptions, joint training without pretraining gets to a lower loss than with pretraining (compare columns iv, v, and vi to i, ii, and iii, respectively). Yet classification accuracy is at chance after joint training without pretraining (columns iii, vii, xi, xv, xix, and xxiii of Table 2). This suggests that joint training without pretraining can memorize the training set yet fail to generalize at all, and further supports claim A and calls claim I into question.

| | | joint | | | no pretraining | | | separate | | |
| | | pretraining | | | | | | | | |
| | | Spampinato et al. (2017) | Li et al. (2021) block | Li et al. (2021) randomized | Spampinato et al. (2017) | Li et al. (2021) block | Li et al. (2021) randomized | Spampinato et al. (2017) | Li et al. (2021) block | Li et al. (2021) randomized |
| | | i | ii | iii | iv | v | vi | vii | viii | ix |
|---|---|---|---|---|---|---|---|---|---|---|
| Inception v3 | i | 1.303 | 1.098 | 0.932 | 1.074 | 0.938 | 0.919 | 1.150 | 1.027 | 1.104 |
| ResNet-101 | ii | 1.155 | 0.925 | 1.019 | 0.963 | 1.006 | 1.080 | 0.885 | 0.936 | 0.884 |
| DenseNet-161 | iii | 1.027 | 1.080 | 1.047 | 0.988 | 0.919 | 0.929 | 1.007 | 0.982 | 1.029 |
| AlexNet | iv | 1.417 | 1.142 | 0.981 | 1.136 | 1.055 | 1.034 | 0.925 | 0.956 | 0.935 |

