# OpenReview forum: "Joint training does not transfer information between EEG and image classifiers"
_ICLR.cc/2026/Conference — Submitted to ICLR 2026_

### Official Review · Reviewer_tdJu · 2025-10-25

**Soundness:** 1
**Presentation:** 1
**Contribution:** 1
**Rating:** 0
**Confidence:** 4

**Summary:**

This paper is very worrisome.

**Ulterior motive**. The introduction attacks Palazzo et al. from the very start. The authors use strong language such as "a large body of flawed work" and "This is further egregious". Many research groups are listed and explicitly attacked in page 3.

**Presentation**. The paper uses way too many of checklist notations:  (i) and (a) in the short abstract, (II) and (B) in the introduction. The introduction starts with a long attack on specific research papers before transitioning with "Here we focus on a separate issue". Page 3 is almost exclusively citations of papers without actual sentences. Page 6 is almost exclusively bullet points.

**Strengths:**

None.

**Weaknesses:**

None.

**Questions:**

None.

**Details Of Ethics Concerns:**

This is clearly a research group settling a dispute in public.

---

### Official Review · Reviewer_QQQ8 · 2025-10-29

**Soundness:** 2
**Presentation:** 1
**Contribution:** 1
**Rating:** 2
**Confidence:** 4

**Summary:**

The paper focuses on an interesting question of whether jointly training the image and the EEG encoder enables the bi-directional information transfer between the two modalities. The author also performed some evaluations with an image-evoked EEG dataset. However, the writing and grammar are very hard to understand. Besides, the analysis is very limited to extensively support the hypothesis. I suggest full improvement after the submission.

**Strengths:**

The main critique in this paper is important in the field of brain decoding. It also raises attention to the fair and strict evaluation in the field.

**Weaknesses:**

N/A

**Questions:**

N/A

---

### Official Review · Reviewer_KJwA · 2025-10-31

**Soundness:** 3
**Presentation:** 2
**Contribution:** 2
**Rating:** 4
**Confidence:** 4

**Summary:**

This paper re-examines the validity of EEG–image joint training approaches originally proposed by Palazzo et al. (2018–2024).

Through PCA variance and classification analyses, they find that pretrained image encoders already contain class information,
and joint training neither enables cross-modal information transfer nor improves performance.
Instead, it degrades image encoder quality, thereby challenging prior claims.

**Strengths:**

- Provides a transparent replication of the EEG–image joint training framework with precise control over confounded and nonconfounded datasets (Li et al., 2021; Spampinato et al., 2017).
- Clearly isolates methodological factors — pretraining, triplet loss, and joint vs. separate training — showing that class information dominates the embedding space.
- The PCA-based decomposition offers an interpretable quantitative view of representational variance and supports the paper’s central logical claims (A–D).
- The writing is coherent and analytical, effectively communicating why observed classification gains stem from class information rather than cross-modal transfer.

**Weaknesses:**

- The PCA-based linear variance analysis may overlook nonlinear dependencies between modalities.
- Variance is used as a proxy for information content without validating its alignment with discriminative information.
- Claims of “class-only” encoding lack information-theoretic validation (e.g., mutual information, RSA).
- Empirical analyses rely on single-subject data (Li et al., 2021) and a pooled dataset (Spampinato et al., 2017), limiting cross-subject generalization.
- The criticism of saliency-map validity is conceptual rather than empirically demonstrated.

**Questions:**

- Q1. Were any analyses performed across multiple subjects or sessions to confirm generalizability?
- Q2. Could additional metrics (e.g., CCA, mutual information) further support the PCA-based conclusions?
- Q3. How did you empirically validate that the saliency/activation maps reflect brain-activity information rather than class priors or ImageNet features?

**Details Of Ethics Concerns:**

Although the accuracy and content of this refutation paper are evaluated independently, there appears to be a concern regarding potential duplicate or salami-sliced submissions.

Specifically, papers #4790 and #4793 present highly similar refutation claims, with Section 2 (Significance) showing nearly identical content. Based on this overlap, it is reasonable to suspect that the same author may have submitted multiple closely related manuscripts.

Moreover, considering reviewer 2p4q’s official comment on paper #4793 and the textual similarity observed in Section 2 of #4802, it seems likely that papers #4790, #4793, and #4802 originate from the same author.

I respectfully ask the ACs and SPCs to review this matter carefully.

---

> ### Author Response · Authors · 2025-11-22
>
> W1: True. But Table 2 and 3, and the associated discussion in lines
> 330-369 are independent of PCA. Further, Table 1, and the associated
> discussion in lines 226-250 provide a lower bound and would only be
> strengthened by potential nonlinear dependencies.
>
> W2: True. But Li et al. (2021), Ahmed et al. (2021, 2022), and
> Bharadwaj (2023), together with submissions 4793, 4796, 4802, and 4804,
> show that there is no descriminative information in the data from
> Spampinato et al. (2017), Kavasidis et al. (2017), Palazzo et
> al. (2018, 2020a, 2020b, 2021, 2024), and 100 other papers.
>
> W3: True. We are not using "information" in the technical sense. The
> paper that we refute, Palazzo et al. (2021), uses the term
> "information" 33 times in an informal sense without any formal
> measurement. We are simply refuting their claims and make no
> information-theoretic claims.
>
> W4: Spampinato et al. (2017), Kavasidis et al. (2017), and Palazzo et
> al. (2018, 2020a, 2020b, 2021, 2024) all present data solely for
> pooled subjects, never cross subject. Our goal here is solely to
> refute their work.
>
> W5: See submission 4796 for an empirical demonstration.

---

### Official Review · Reviewer_KcsM · 2025-11-10

**Soundness:** 2
**Presentation:** 1
**Contribution:** 2
**Rating:** 2
**Confidence:** 2

**Summary:**

This paper reexamines a previously proposed method for jointly training EEG and image encoders and claims that it does not achieve meaningful information transfer between modalities. Using several architectures and datasets, the authors claim that the joint model’s performance is largely driven by preexisting class information rather than shared brain–image representations. Their analyses suggest that such training can even degrade image-model performance and fails to generalize on properly controlled data. The work provides a methodological critique of a widely cited line of research.

**Strengths:**

* the work performs a large number of analyses to investigate different claims made in prior work in a variety of settings
* it is important to critically evaluate and correct where necessary any mistaken approaches present in the literature

**Weaknesses:**

I found the writing hard to follow:
* it lacks a clear structure, unnecessarily repeating  the same content (e.g., Introduction: "Other work questioned these claims due to confounds...", same point again in 2 ("Several independent lines have refuted [...] suffer from temporal confound..."))
* using up almost a whole page for lists of citations
* mixes logical argumentation with general arguments on the importance of refutation or the importance of the release of raw data in distracting ways
* Long bulleted lists without clear internal structure with long texts inside each bullet point

Some of the space would be better used to more clearly convey the motivation behind some analysis choices or illustrative figures explaining the analysis pipelines. In the current form, I do not find it suitable for publication as I do not believe it efficiently communicates the claims it tries to make.

Overall, I have to admit I still don't really get the point. If one wants to check whether an EEG encoder and an image encoder have had some information transfer, I would expect it should be enough to
1) Learn some matching of EEG and image encodings
2) Evaluate whether unseen unconfounded EEG and corresponding image encodings are mapped closer together than non-corresponding ones. Images could also be of classes not present in the training set if one wants to remove more confounds.

Am I misundersanding that? If that had already been done in previous work, I do not see the point of deep analysis of in how far (only) class information is learned or not.

Especially as such analyses are hard:
* PCA analysis may miss nonlinear relationships obviously
* "Prior to joint training, the pretrained image encoders contain close to perfect class information, and likely very little information other than class information." -> The second part seems at odds with prior literature that was able to invert a lot of semantic details from images even when just using output logits of pretrained image encoders.
* "have the representational capacity to memorize one-hot EEG and image encodings that minimize the loss function on the training set." -> We know that there are many settings in which deep networks have the capacity for memorization yet learn generalizable representations, so this does not seem such a strong argument to me

**Questions:**

I do not understand the top-40 components analysis and the motivation behind it? Because there are 40 classes?

---

> ### Author Response · Authors · 2025-11-22
>
> Re "the point": We are not claiming that it is impossible to jointly
> train EEG and image classifiers to produce common embeddings. We are
> solely claiming that the method of Palazzo et al (2021) does not do
> so.  Palazzo et al. (2021) claim that "our multimodal approach learns
> a joint brain-visual embedding and finds similarities between brain
> representations and visual features." Palazzo et al. (2021) never
> perform step (2) that the reviewer suggests. We can't refute a
> procedure that Palazzo et al. (2021) never perform. The sole evidence
> given in Palazzo et al. (2021) is Tables 3 and 4. The present
> manuscript refutes those claims.
>
> See reponse to KJwA wrt nonlinear relationships.
>
> Re "at odds": We make no claim about any other work in the
> literature. Our claims solely refute Palazzo et al. (2021). While
> other work in the literature may be "able to invert a lot of semantic
> details from images even when just using output logits of pretrained
> image encoders", our analyses provide strong evidence that for the
> particular pretrained image encoders used in Palazzo et al. (2021) on
> the particular stimuli and EEG rcordings used with the particular
> training regimen used on the particular classification problems being
> evaluated, there is very little information other than class
> information in the pretraind image encoders.
>
> Re "representational capacity": Our whole point about memorization is
> not that models that memorize can't generalize. We make no claim about
> generalization. Our sole goal is to refute the claim by Palazzo et
> al. (2021) that joint training transfers information. We show that
> independent training can arrive at the same point in model space as
> joint training. And  actually does arrive at a better point. So joint
> training with the specific loss function used by Palazzo et al. (2021)
> doesn't accomplish anything that couldn't be accomplished with
> independent training.
>
> Q: Yes, top 40 component because of 40 classes. The pretrained Image
> encoders of Palazzo et al. (2021) were trained with one-hot class
> labels and the pretrained Image encoders were used to jointly train
> the EEG encoders, implicitly training the whose assembly with one-hot
> class labels. The whole purpose of doing this analysis is to refute
> the claim in Palazzo et al. (2021) "Note that class labels are
> not used anywhere in the equation." by showing that class information
> and only class information is used during joint training.

---

### Author Response · Authors · 2025-11-22
**Historical Background and Significance**

To understand this work's significance, consider this brief historical
overview.

Spampinato et al. (2017) introduced a block-designed dataset
("Perceive") and methods that claim to achieve extremely high accuracy
decoding stimulus image class from EEG recordings. This was amplified
by follow on papers (Kavasidis et al. 2017, Palazzo et al. 2018,
2020a, 2020b, 2021), many of which claim to do things like reconstruct
stimulus images from EEG recordings. Further, Tirupattur (2018) does
this with a fresh dataset (Kumar 2018) that has the same block design.

Li et al. (2021) debunked all of the above, demonstrating that the
Perceive dataset suffers from a block confound. EEG exhibits drift,
encoding a clock in the signal. Since Perceive was collected with all
and only stimuli of the same class being temporally adjacent, the
classifier can mistakenly classify the clock/drift instead of the
stimulus-related EEG response. Follow on papers (Ahmed et al. 2021,
2022, Bharadwaj et al. 2023) added novel independent confirmation of
the results of Li et al. (2021).

Despite this, Palazzo et al. (2020b, 2021, 2024) continue to argue
that their dataset is valid. At this point, there are over one hundred
papers that use the Perceive dataset, the Kumar (2018) dataset, or
other datasets that suffer from the same block confound. Many new
datasets have been collected with this same block confound, some of
which are becoming widely used. The vast majority of these were
published after the confound became known (Li et al. 2021). Some of
these are unaware of the confound. Others are aware, but dismiss it,
often based on the argument of Palazzo et al. (2020b, 2021, 2024).

That argument is what this manuscript refutes.

This confound has been extensively debated on blogs like reddit, yet
that too has not stopped the extensive publication of flawed work.

There are three distinct levels of severity of this issue, which
progressively support greater need for continued publication:

 1. Many authors are unaware of the confound, despite the fact that it
    was published in prominent venues (e.g., TPAMI, CVPR, NeurIPS) and
    continue to publish flawed work

 2. While many authors are aware of the confound, they nonetheless
    ignore the warning and continue to publish flawed work.

 3. Some authors dismiss the confound and actively argue for the
    community to continue to employ flawed methods.

---

> ### Author Response · Authors · 2025-11-22
> **Re: Concerns about duplicate submissions**
>
> We submitted five manuscripts to ICLR 2026. To summarize:
>
>   4790: Palazzo (2020b, 2021) introduces a method that claims to
>         jointly train two mappings, from EEG and images, to a common
>         embedding space. We debunk central claims about this
>         embedding. We do this both for the confounded dataset and
>         nonconfounded datasets.
>
>   4793: Palazzo et al. (2020b, 2021, 2024) claim that their dataset
>         does not suffer from drift. We show that three other datasets,
>         all collected by different people in different labs, suffer
>         from drift, demonstrating that drift is unavoidable with EEG.
>
>   4796: Palazzo et al. (2020b, 2021) produce activation maps and claim
>         that these are consistent with neuroscience knowledge. We
>         debunk this claim.
>
>   4802: Palazzo et al. (2024) further claim that their dataset is
>         valid by arguing that the experiments in Li et al. (2021),
>         Ahmed et al. (2021, 2022), and Bharadwaj (2023) were
>         improperly conducted. We repeat the experiment in Spampinato
>         et al. (2017) exactly, in a controlled fashion, where the only
>         thing varied is block order. this conclusively demonstrates
>         that Spampinato et al. (2017), Kavasidis et al. (2017),
>         Palazzo et al. (2018, 2020a, 2020b, 2021, 2024), and the one
>         hundred other papers are flawed.
>
>   4804: Palazzo et al. (2024) makes numerous false statements about
>         Li et al. (2021), Ahmed et al. (2021, 2022), and Bharadwaj (2023).
>         We correct those statements.
>
> These are all independent. There is no duplicate substantive material
> between these five submissions and Li et al. (2021), Ahmed et al.
> (2021, 2022), and Bharadwaj (2023). While they all comment on the same
> body of flawed work, they each introduce and discuss distinct
> technical issues and make distinct contributions.
>
> We included §2 Significance in all five manuscript. While largely the
> same text, this is not the technical contribution of each respective
> manuscript. It solely serves to highlight the significance of the
> specific technical contribution in each individual manuscript, namely
> that each offers independent refutation of one hundred papers. This is
> important, because even if one were to remedy one of the flaws, many
> others remain, and a large and growing corpus of work remains flawed.
> Further, it is conceivable that in the future, a paper might suffer
> from one flaw but not the other, yet it would still be invalid.

---

> > ### Author Response · Authors · 2025-11-22
> > **Re: Public debate**
> >
> > Several reviewers commented that public debate of this issue is
> > inappropriate. We realize that this may be unconventional and uncommon
> > in the ML community. But it is common in most other scientific fields
> > (e.g., Brain and Behavioral Science, Psycoloquy, ...). Public debate
> > through publication is the well-established method for arriving at
> > scientific truth. Schaeffer (2025) have argued that a mechanism for
> > publishing critiques and refutations is sorely lacking in ML.
> >
> > The vast majority of the reviews of all five of these submissions
> > focus on the fact that they are unconventional. Essentially none of
> > the reviews discuss any technical flaws in these submissions. We
> > would be happy to discuss and address any technical flaws.

---

> > > ### Author Response · Authors · 2025-11-22
> > > **Specific relevance and significance to ICLR and the ML community**
> > >
> > > It is important, if not imperative, for the community to publish this
> > > work. Without it, the community continues to submit and publish more
> > > flawed work at a growing rate. Fifty new papers papers have been
> > > published since the flaw was first reported in prominent venues: once
> > > in CVPR (Ahmed et al. 2021) and three times in TPAMI (Li at al. 2021,
> > > Ahmed et al. 2022, Bharadwaj et al 2023).
> > >
> > > Some recent flawed work has been published even by the ML community in
> > > top ML venues, despite awareness of the issue: Liu et al. (2024) in
> > > NeurIPS collects a new dataset that suffers from the block
> > > confound. While the authors cite Li et al. (2021) and Ahmed et al
> > > (2021), they fail to appreciate (or maybe hide the fact) that their
> > > work is confounded.
> > >
> > > Some recent flawed work has even been submitted to ICLR 2025 (and
> > > apparently resubmitted to ICLR 2026 despite reviewer warnings). It
> > > appears that even the reviewer pool of ICLR is unaware of the severity
> > > of the confound.
> > >
> > > https://openreview.net/forum?id=ejVuTFFkl6&noteId=zafmRtlFw1
> > >
> > > collects a new dataset that suffers from the block confound. While the
> > > authors again cite Li et al. (2021), they incorrectly claim that their
> > > dataset does not suffer from the confound. All four of the reviewers
> > > point this out. While this submission was rejected, three of the
> > > reviewers rated it as "Soundness: 3: good" and two of the reviewers
> > > rated it as "Contribution: 3: good".
> > >
> > > The apparent resubmission (18265) to ICLR 2026 again cites Li et
> > > al. (2021) and again incorrectly claims that their dataset does not
> > > suffer from the confound. Again, three of the four reviewers point out
> > > that this work suffers from the block confound. And again, two of the
> > > reviewers rate this as "Soundness: 3: good", one of the reviewers
> > > rates this as "Contribution: 3: good", and one even rates this as
> > > "Contribution: 4: excellent" and recommends acceptance.
> > >
> > > We have a simple question for the reviewers, area chairs, and program
> > > chairs: If one cannot publish refutations like this in ICLR, how else
> > > do you propose we address the fact that there is a large and growing
> > > body of flawed work being published?

---

### Meta-Review · Area_Chair_HWQJ · 2026-01-12

**Summary:**

The submitted manuscript appears to be a review paper.

I think that `KcsM`'s final comment and the author response is perhaps most informative: this manuscript seems to be mostly about how the method of Palazzo et al. (2018, onward) does not produce common EEG/Image embeddings, paraphrased from author response.

In my opinion, if the belief that a large branch of EEG embedding literature is compromised by bad scientific practice, it would be most constructive to demonstrate so empirically (admittedly done so in section 4 of the submission), and to do so in a way that is unquestionably clear. I tend to agree with the comments of `KJwA`, that the use of PCA variance in top 40 components is necessarily limiting.

However, as the authors strongly believe there to be flaws in Palazzo et al. sequence of works, it feels as though there must be more direct demonstrations of this fact. I also recommend featuring this graphically, in section 1 of the paper. Both `KcsM` and `KJwA` appear to have waded through the entire manuscript. I broadly agree with their comments: the result and conclusion should be immediate, not a search through the text. Arguably the authors OpenReview posts have been more clear; these then should be the main text.

**Reviewer Concerns:**

Reviewer concerns of `KcsM` and `KJwA` are not sufficiently addressed. I have disregarded the other two reviews.

**Reviewer Scores:**

None

---

### Decision · Program_Chairs · 2026-01-26

Reject